



# FLOWERS: An integral approach to engineering wake models

Michael J. LoCascio[1,2], Christopher J. Bay[1], Majid Bastankhah[3], Garrett E. Barter[1], Paul A. Fleming[1], and Luis A. Martínez-Tossas[1]

[1]National Wind Technology Center, National Renewable Energy Laboratory, Golden, CO 80401, USA
[2]Civil and Environmental Engineering, Stanford University, Stanford, CA 94305, USA
[3]Department of Engineering, Durham University, Durham DH1 3LE, UK

**Correspondence:** Luis A. Martínez Tossas (luis.martinez@nrel.gov)

**Abstract.**

Annual energy production (AEP) is often the objective function in wind plant layout optimization studies. The conventional method to compute AEP for a wind farm is to first evaluate power production for each wind direction and speed using either computational fluid dynamics simulations or engineering wake models. The AEP is then calculated by weighted-averaging (based on the wind rose at the wind farm site) the power produced across all wind directions. We propose a novel formulation for time-averaged wake velocity that incorporates an analytical integral of a wake deficit model across every wind direction. This approach computes the average flow field more efficiently, and layout optimization is an obvious application to exploit this benefit. The clear advantage of this new approach is that the layout optimization produces solutions with comparable AEP performance yet is completed about 700 times faster. The analytical integral and the use of a Fourier expansion to express the wind speed and wind direction frequency create a more smooth solution space for the gradient-based optimizer to excel compared with the discrete nature of the existing weighted-averaging power calculation.

## 1 Introduction

The layout of a wind plant is a primary design element that influences its performance. Optimizing the layout can be thought of as a wake avoidance problem, wherein turbines are placed such that they avoid the wakes from other turbines as much as possible. Power losses from wake interactions can be on the order of 10–20% in wind farms (Barthelmie et al., 2007, 2009). When turbines are placed within about 3 rotor diameters, these power losses can be as high as 40% (Stanley et al., 2019); even within 15 diameters, wake interactions are non-negligible (Meyers and Meneveau, 2012).

In controls and optimization applications, the wake velocity deficit is approximated with low-fidelity analytical models. The classical tophat model parameterizes the wake expansion rate and computes the wake deficit as a function of downstream position (Jensen, 1983). Improvements on this approach aim to replace the discrete boundaries of the tophat model with a continuous profile, such as the Jensen-cosine (Tian et al., 2015, 2017) and Gaussian (Bastankhah and Porté-Agel, 2014) models. More involved engineering models such as the TurbOPark model (Nygaard et al., 2020) account for wake combination and wake-added turbulence more formally in their formulation. The curled wake model is a mid-fidelity numerical model derived from the Reynolds-averaged Navier-Stokes equations (Martínez-Tossas et al., 2019, 2021). The trade off to explicitly capturing



more of the flow physics is the added complexity, both in the calibration of additional parameters and in computational cost. These steady-state wake models are well-suited to estimate wake velocity in simulations with a single wind direction. However, computing average wake velocities or energy production for different wind speeds and directions requires averaging the results of multiple simulations. This process is cumbersome, especially with the more complex models like the curled wake model.

Layout optimization studies leverage these low-fidelity models to approximate the wake velocity within the wind farm. Tur-
bines are placed to minimize wake interactions and thereby maximize annual energy production (AEP) of the plant. Gradient-free algorithms are common practice in industry for small wind farms (Stanley et al., 2021), but scale poorly with additional degrees of freedom (Herbert-Acero et al., 2014; Ning and Petch, 2016). Gradient-based optimization, on the other hand, is more robust in systems with a larger number of design variables. The simplest structure for the design variables is to assign the position of each turbine independently (Feng and Shen, 2015; Gebraad et al., 2017). A strategy to reduce the number of
design variables is to restrict the layout to a grid (González et al., 2017; Perez-Moreno et al., 2018), or use a combination of placement along the farm boundary and a grid on the interior (Stanley and Ning, 2019). These approaches reduce the cost of the layout optimization study, especially for larger wind farms. However, they restrict the freedom and flexibility of the wind farm developer and produce simplistic layouts that can underperform in practice. Research that addresses the calculation of AEP has focused on statistical methods to improve the efficiency of estimating this quantity (King et al., 2020; Padrón et al.,
2019). These data-driven approaches are promising but leave room for a physics-based framework to modify the formulation of AEP.

AEP is an integral quantity. The total power production of a wind plant is calculated based on the wind speed flowing through each turbine. For a single wind speed and direction, this procedure is straightforward. Figure 1 illustrates the flow field around a single turbine, as is often studied in wake modeling problems. The velocity contour plots represent the weighted-averaged flow
distribution based on the wind roses shown below each respective contour plot. Figure 1a shows the velocity distribution when the wind rose contains only one non-zero wind direction. If there is more than one wind direction, the flow fields from each speed-direction bin must be averaged with weights equal to their normalized frequency as seen in Figure 1b-c. For example, in Figure 1b the frequency of the 180° wind direction is greater than that for 225° (and the freestream wind speed is held constant), and so the velocity contour plot shows the wake directed horizontally with a stronger velocity deficit compared with
the angled wake. This procedure is extended across every discrete wind speed-direction bin with the contribution to the sum weighed by the frequency of that bin. Figure 1c illustrates the averaged flow field and how the higher-frequency wind directions manifest as more pronounced wake deficits in the contour plot. The AEP of the wind plant is therefore a numerical integral of the power as a function of wind speed and direction.

The inspiration for the FLOW Estimation and Rose Superposition (FLOWERS) flow field model is to analytically compute
the average wake velocity given the frequency and magnitude of the wind speed coming from every direction. Since the average wake velocity is conceptualized similarly to AEP, extending the FLOWERS approach to calculating AEP would be straightforward. We hypothesize that the analytical integration will considerably reduce the computational cost of average wake velocity and AEP calculations compared to the numerical integration.

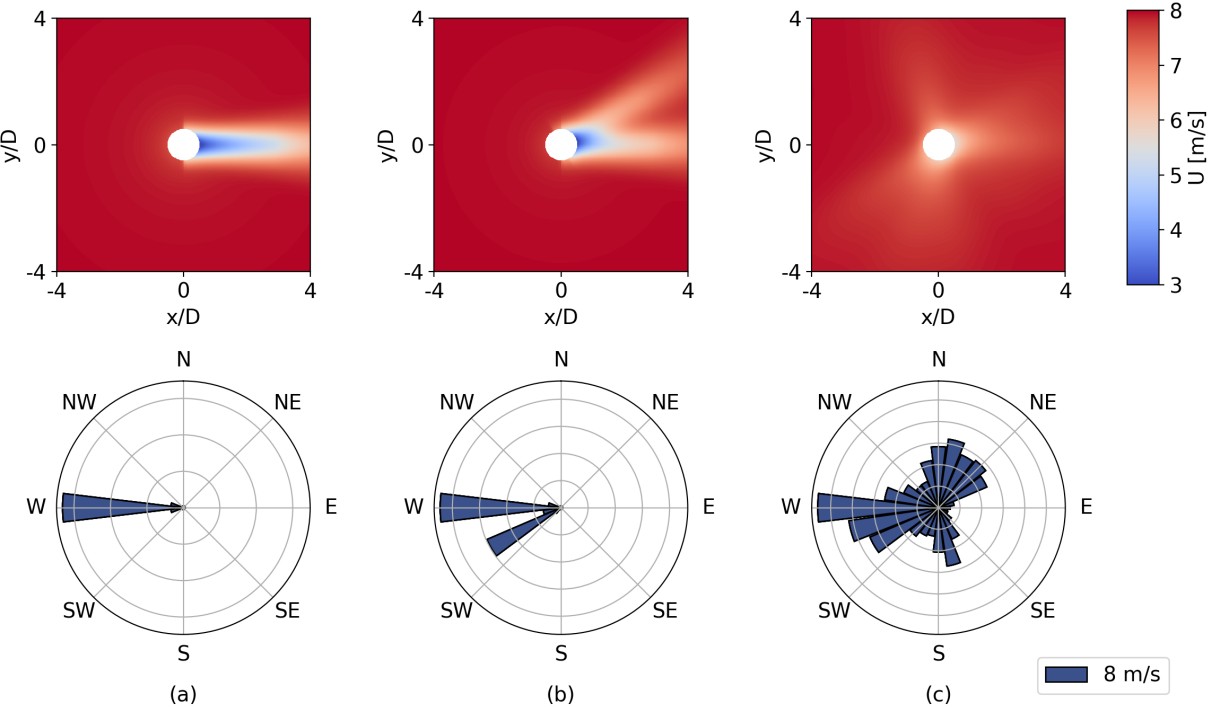

**Figure 1.** Left: velocity contour plot of flow through a wind turbine for a single wind direction. Center: averaging effect of two wind directions. Right: the annually-averaged velocity flow field. Note that the wind roses (bottom) display the frequency of each wind direction with a constant wind speed of 8 m/s for every direction.

In this paper, we first derive the equations for the time-averaged wake velocity and a new formulation for AEP in Section 2, including its application in the wind plant layout optimization problem. In Section 3, the AEP calculations from FLOWERS are compared to the conventional numerical integral method, which is the standard method in the NREL FLORIS model (NREL, 2021). Finally, in Section 4, the performance of the FLOWERS AEP in the optimization of wind farm layouts is compared against the traditional optimizer.

## 2 Mathematical Formulation

### 2.1 Time-Averaged Wake Speed

To derive a mathematical formulation for the time-averaged flow distribution, we use the classical Jensen (tophat) wake deficit model (Jensen, 1983):

$$\frac{U(x,y)}{U_\infty} = 1 - \frac{1 - \sqrt{1 - C_T}}{(kx+1)^2} W(x,y), \tag{1}$$



where $x$ and $y$ are the streamwise and spanwise position, respectively, normalized by the rotor radius. We only consider the

2D plane at hub height such that wake deficit is not a function of a vertical position $z$. $C_T$ is the thrust coefficient which is assumed to be constant for simplicity, $k$ is the wake expansion coefficient, $U$ is the wake speed and $U_\infty$ is the freestream wind speed. $W(x,y)$ represents the Jensen wake region: $W(x,y) = 1$ if $|y| \leq kx + 1$ and is zero elsewhere.

We transform Eq. 1 from Cartesian to polar coordinates denoted by $r$ and $\theta$, where $r = \sqrt{x^2 + y^2}$ and $\tan(\theta) = y/x$. We allow the wind direction $\theta'$ to be variable in Eq. 2:

$$U(r,\theta,\theta') = U_\infty(\theta')\left[1 - \frac{1 - \sqrt{1 - C_T}}{(kr\cos(\theta - \theta') + 1)^2}\, W(r\cos(\theta - \theta'), r\sin(\theta - \theta'))\right]. \tag{2}$$

To clarify, $\theta$ is the angular position in polar coordinates where we wish to compute the average wake velocity, and $\theta'$ is the wind direction defined within that coordinate frame. We integrate across all wind directions $\theta'$ to compute the average wake speed at a given location $(r,\theta)$. In doing so, we weight each wind direction with its frequency $f(\theta')$, so the weighted-averaged wind speed denoted by $\overline{U(r,\theta)}$ is written as:

$$\overline{U(r,\theta)} = \frac{1}{2\pi}\int_{-\pi}^{\pi} U_\infty(\theta')f(\theta')\left[1 - \frac{1 - \sqrt{1 - C_T}}{(kr\cos(\theta - \theta') + 1)^2}\, W(r,\theta,\theta')\right]d\theta'. \tag{3}$$

We define two new variables to simplify this expression. Let $u = \theta - \theta'$ be the angular position relative to the wind direction. Also, the wake region $W(r,\theta,\theta')$ is zero for $\theta$ outside of the wake region defined by $\theta_c$: $\theta = \theta' \pm \theta_c$. In our coordinate system, the wake geometry is defined by the line $\sin(\theta_c) = k\cos(\theta_c) + 1/r$. We solve this equation for $r > 1$ since we are interested in the wake speed at positions outside of the rotor area:

$$\tan(\theta_c) = \frac{\frac{1}{r} + k\sqrt{1 + k^2 - (\frac{1}{r})^2}}{-\frac{k}{r} + \sqrt{1 + k^2 - (\frac{1}{r})^2}}. \tag{4}$$

We must include the interaction between multiple wakes for this formulation to be useful in systems of several turbines. The wake velocity is defined as the difference between the freestream velocity and the velocity deficit. Eq. 3 can be split into these two components. Then, we linearly superimpose the deficits to form a relation for the total velocity deficit caused by all turbines:

$$\overline{U(r,\theta)} = \frac{1}{2\pi}\int_{-\pi}^{\pi} U_\infty(\theta')f(\theta')\,d\theta' - \frac{1}{2\pi}\sum_i \int_{-\theta_c}^{\theta_c} U_\infty(\theta_i - u)f(\theta_i - u)\left[\frac{1 - \sqrt{1 - C_T}}{(kr_i\cos(u) + 1)^2}\right]du, \tag{5}$$

where $r_i$ and $\theta_i$ are the relative radius and polar angle with respect to each turbine position ($r_i = \sqrt{(x - x_i)^2 + (y - y_i)^2}$ and $\tan(\theta_i) = y_i/x_i$, where $x_i$ and $y_i$ represent the position of the center of the *ith* turbine). The relevant information about the wind conditions is the wind direction $\theta'$, the wind speed $U_\infty(\theta')$, and the wind direction frequency $f(\theta')$. Note that for simplicity a single average wind speed is used for each wind direction. These quantities are specified for a particular location

by the wind rose. In practice, the wind rose is a discrete data set in which wind directions (and their associated average speeds





and frequencies) are binned. The product $U_\infty(\theta')f(\theta')$ is a vector with length equal to the number of wind direction bins. We define $g(\theta') = \frac{1}{2\pi}U_\infty(\theta')f(\theta')$. If we expand $g(\theta')$ with a Fourier series:

$$g(\theta') = \frac{a_0}{2} + \sum_{n=1}^{N} a_n \cos(n\theta') + b_n \sin(n\theta'), \tag{6}$$

the wind rose is defined continuously rather than discretely, where $a_0$, $a_n$ and $b_n$ are Fourier coefficients and can be easily

found for a given $g(\theta')$. For a wind rose with $B$ wind direction bins, the maximum number of terms in this discrete Fourier transform is $N = \mathrm{ceiling}(B/2)$, where "ceiling" indicates that we round up to the nearest integer. Also, we approximate the fraction in the second term in the right hand side of Eq. 5 using a Taylor series:

$$\overline{U(r,\theta)} = \int_{-\pi}^{\pi} g(\theta')\, d\theta' \; - \left[1 - \sqrt{1-C_T}\right] \sum_i \int_{-\theta_c}^{\theta_c} g(\theta_i - u)\left[\frac{1}{(kr+1)^2} + \frac{kru^2}{(kr+1)^3}\right] du. \tag{7}$$

The first integral in the right-hand side of Eq. 7 represents the weighted-average of the freestream velocity, denoted by $\overline{U_\infty}$

hereafter. Using Eq. 6,

$$\overline{U_\infty} = \int_{-\pi}^{\pi} g(\theta')\, d\theta' = \int_{-\pi}^{\pi}\left[\frac{a_0}{2} + \sum_{n=1}^{N} a_n \cos(n\theta') + b_n \sin(n\theta')\right] d\theta' = a_0\pi. \tag{8}$$

The second integral on the right-hand side of Eq. 7 represents the average wake velocity deficit $\overline{\Delta U_i(r_i, \theta_i)}$. Using Eq. 6,

$$\overline{\Delta U_i(r_i,\theta_i)} = \left[1 - \sqrt{1-C_T}\right]\int_{-\theta_c}^{\theta_c}\left[\frac{a_0}{2} + \sum_{n=1}^{N} a_n \cos(n(\theta_i - u)) + b_n \sin(n(\theta_i - u))\right]\left[\frac{1}{(kr_i+1)^2} + \frac{kr_i u^2}{(kr_i+1)^3}\right] du.$$

Solving the above integral yields:


$$\overline{\Delta U_i(r_i,\theta_i)} = \left[1 - \sqrt{1-C_T}\right]\left[\frac{a_0\theta_c[kr_i(\theta_c^2 + 3) + 3]}{3(kr_i+1)^3} + \right.$$
$$\left. \sum_{n=1}^{N} \frac{2[a_n\cos(n\theta_i) + b_n\sin(n\theta_i)]}{[n(kr_i+1)]^3}\left[\sin(n\theta_c)\{n^2[kr_i(\theta_c^2+1)+1] - 2kr_i\} + 2n\theta_c kr_i\cos(n\theta_c)\right]\right]. \tag{9}$$

The time-averaged wake speed $\overline{U(r,\theta)}$ at a given location $(r,\theta)$ is the difference between these two terms:

$$\overline{U(r,\theta)} = \overline{U_\infty} - \sum_i \overline{\Delta U_i(r_i,\theta_i)}, \tag{10}$$

where $\overline{U_\infty}$ and $\overline{\Delta U_i(r_i,\theta_i)}$ can be found from Eq. (8) and Eq. (9), respectively.

## 2.2 Annual Energy Production

The power $P$ produced by a turbine is a function of the incoming wind speed:

$$P(U) = \frac{1}{2}C_p(U)\rho A U^3, \tag{11}$$





where $\rho$ is the density of air (assumed to be constant), $C_p$ is the power coefficient, $A$ is the swept area of the rotor, and

$U$ is the velocity at the location of the center of the turbine when it is not present. The average turbine power would be $\overline{P(U)} = \frac{1}{2}\rho A \overline{C_p(U)U^3}$. However, the integral to compute this average is intractable in our formulation. The power coefficient is dependent on the local wind speed, but $U$ is not known as a function of wind direction prior to the integration; to obtain it, we would need to calculate the wake speed independently for each wind direction, which negates the purpose of the FLOWERS approach. Also, including the nonlinear $U^3$ term in the integral introduces complications with the wake superposition and the

definition of the independent wake regions. We make two simplifications to compensate: $C_p$ is calibrated as a function of the average wake speed $\overline{U}$, and we substitute $\overline{U}^3$ into Eq. 11 instead of evaluating $\overline{U^3}$. Therefore, the AEP for a given turbine is proportional to $C_p(\overline{U})\overline{U}^3$.

### 2.3 Layout Optimization Problem

We apply this novel formulation of time-averaged wake velocity and AEP to the wind plant layout optimization problem. For

$M$ turbines in a wind plant, there are $2M$ design variables to independently specify the position of each turbine in 2D space. The objective function is the total AEP of the plant, which we aim to maximize. The turbines are constrained within a specified boundary with a minimum separation of two rotor diameters. Gradient-based optimization is performed with the Sequential Least Squares Programming (SLSQP) algorithm. This optimizer solves a minimization problem, so we technically minimize $-AEP$:

$$\min_{x_i,y_i} \quad -AEP(x_i,y_i,\theta',U_\infty(\theta'),f(\theta'))$$

$$\text{s.t.} \quad \text{boundary constraints} \tag{12}$$

$$S_{ij} \geq 2D,$$

where $x_i$ and $y_i$ are the center of the turbine $i$ in Cartesian coordinates and $S_{ij}$ represents the separation between the centers of turbines $i$ and $j$.

  The primary benefits of FLOWERS lie in its suitability to drive layout optimization as a wake avoidance problem, despite the fact that simplifications made to develop FLOWERS might induce some errors in the predicted magnitude of AEP. The

optimizer relies on the objective function to provide a quantitative metric to compare possible solutions; in this case with gradient-based optimization, the ratio of the objective function evaluated for two different solutions is of importance. In other words, the objective function's output itself is not necessarily critical as long as the mapping between inputs and outputs in the function remains consistent. If we think of the layout optimization problem as a wake avoidance problem, then the objective function must be able to approximate wake magnitudes and downstream influence to minimize their interactions. Turbines

aligned with predominant wind directions and turbines with close spacing will reduce AEP in the FLOWERS optimization, just as it will in the conventional optimization. Wake avoidance can be achieved despite a less accurate estimate of AEP because the factors that cause positive or negative changes to AEP are still present. The gradient throughout the optimization space will be different because the objective functions are not identical. However, with a sufficiently strict convergence criterion, we





predict that the FLOWERS optimization will still find a similar quality solution to the standard technique. A more accurate
AEP estimate can be added as a final post-processing step once the layout optimization is complete.

## 3 AEP Comparison

### 3.1 Aligned Case

We start by comparing the AEP estimates for an illustrative test case of three turbines aligned with a predominant wind
direction. The AEP for the conventional approach is computed using the Jensen wake deficit model with the same wake
expansion coefficient as in FLOWERS ($k = 0.05$). We prescribe constant $C_p = 0.4365$ and $C_T = 0.75$. The rotor diameter is
$D = 126$ m throughout this paper based on the NREL 5MW Jonkman et al. (2009). Figure 2 illustrates the wind rose and farm
layout for this case and the flow fields generated from the FLOWERS and Jensen models.

The FLOWERS AEP is 1.8% lower than the result from the numerical integration approach. Substantial wakes only exist
for three discrete wind directions in this example, so the profile of $U$ versus $\theta'$ is mostly uniform except for a sharp decrease in
these aligned orientations. As a result, the difference between $\overline{U}^3$ and $\overline{U^3}$ contributes to this small discrepancy. Also, the aligned
placement of the turbines leads to strong wake deficits. In FLOWERS, we neglect local wind conditions when computing the
wake velocity and instead normalize the velocity deficit by the global freestream velocity and assume a constant $C_p$ across
all turbines. When the magnitude of the wake velocity deficit is pronounced, these assumptions in the FLOWERS formulation
become more noticeable. Despite the differences in modeling assumptions, the AEP estimates match closely.

One important feature of the FLOWERS solution is its smoothness. Despite using the discrete tophat model, the flow field in
Figure 2b is continuous. The Fourier transform of the wind rose information and the analytical integration results in a smooth
solution. On the other hand, the numerical integration in the conventional approach relies on discrete wind direction bins. The
contour plot in Figure 2c clearly illustrates the discrete boundaries of the wakes for the three dominant wind directions.

### 3.2 Generalized Case

We now examine the differences in AEP between FLOWERS and the numerical integration approach more broadly. Forty
randomized test cases were generated. A random number of turbines between 4 and 50 was chosen for each. The layout of the
turbines is randomized within a square boundary of side length $25D$, and a minimum separation of $4D$ between the turbines is
enforced. Each wind rose is randomly selected from the WIND Toolkit (Draxl et al., 2015). Figure 3 displays the number of
turbines and annual average wind speed for each case.

We compare the computation time and percent difference in AEP between the two methods in Figure 4. The FLOWERS
AEP computation is on average about 22 times faster than the Jensen model. This difference scales with the size of the wind
farm, in part because FLOWERS is only computing the velocity through each turbine at a single point instead of an array of
points on the rotor area.





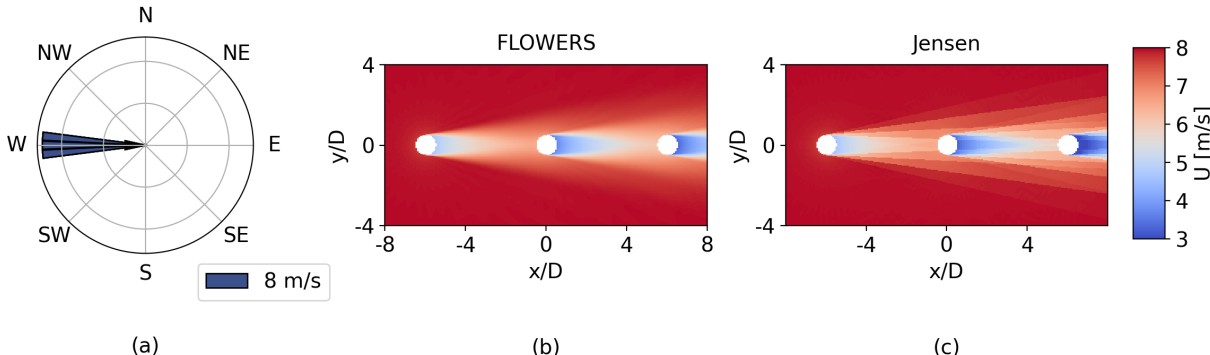

(a)         (b)         (c)

**Figure 2.** AEP comparison for three turbines aligned with the predominant wind direction with $6D$ spacing. The freestream wind speed is a constant 8 m/s. The number of wind direction bins $B$ used in (c) is $B = 72$, and the number of Fourier coefficients used to plot results in (b) is $N = 37$.

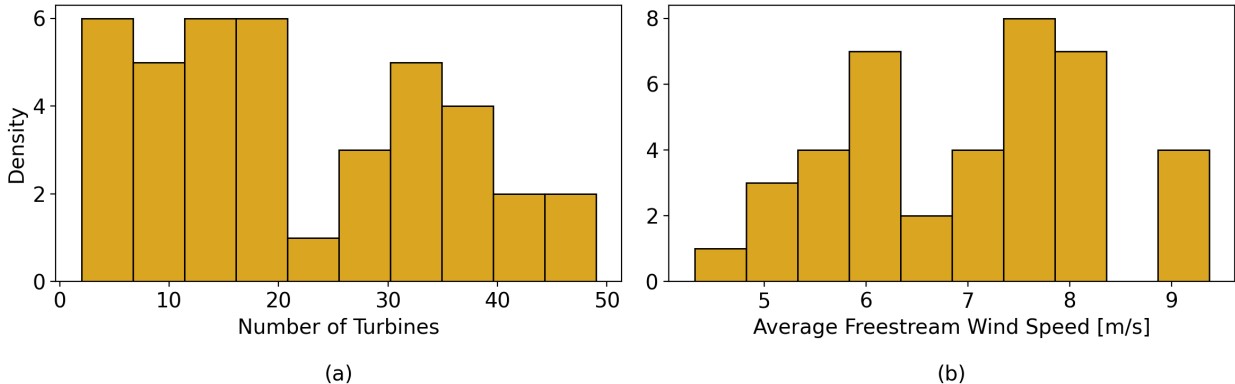

(a)                            (b)

**Figure 3.** 40 test cases with random layouts and random wind roses to compare AEP more generally.

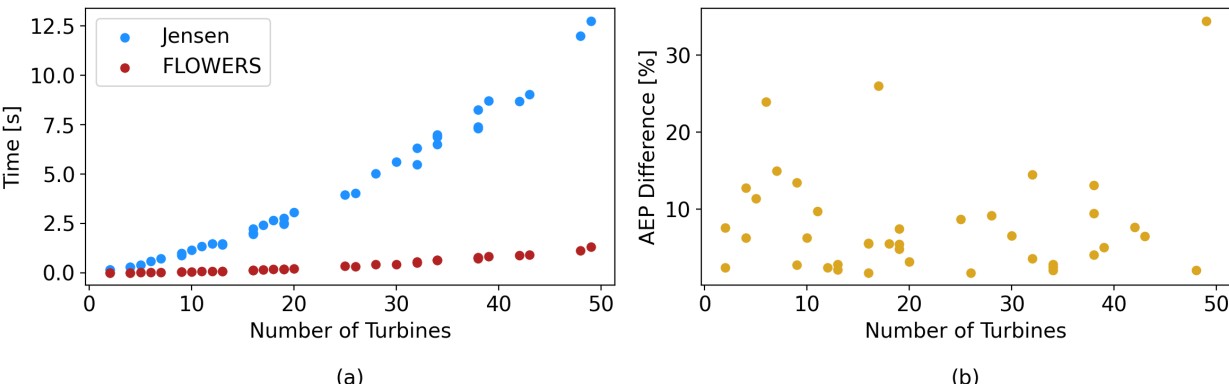

**Figure 4.** Comparison of computational cost and relative difference in AEP between FLOWERS and Jensen for the randomized cases.

The discrepancy in AEP between the two methods is more pronounced in these randomized cases. The freestream wind speed
is not held constant here as it was in the first example. More variations in $U$ across different wind directions results in more
error between the two methods due to the approximations built into the FLOWERS formulation. The common characteristics
of the cases with percent difference greater than 20% are a freestream wind speed consistently less than 5 m/s and a small
number of predominant wind directions. The average difference between FLOWERS and the Jensen integration is 8%.

This difference in AEP between the two methods is not necessarily a fatal flaw. FLOWERS is likely not a reliable prediction
of AEP for a wind farm, but it is difficult to expect a highly accurate and precise estimate of AEP from a low-fidelity wake
model anyway. However, as we will illustrate shortly, it is still possible to use the FLOWERS AEP in the layout optimization
problem. In fact, the FLOWERS AEP calculation is better suited for layout optimization problems than the conventional
method. More precise AEP estimates can always be generated as a final post-processing step after the layout optimization is
complete.

## 3.3 Improving Computational Efficiency

Before addressing the utility of the FLOWERS formulation in the layout optimization problem, we can explore how to further
improve the computational time. For the conventional numerical integration method, a common approach to reduce the cost of
calculating AEP is to reduce the number of wind speed-direction bins, thereby reducing the number of simulations that must
be run. Figure 5a illustrates how the computational time of the AEP calculation is roughly proportional to the number of bins.
Each bin adds an identical set of function calls and operations to the summing process, so the cost scales linearly. The data
presented here is for a 7x3 grid of turbines with $5D$ spacing in all directions and five wind roses randomly sampled from the
WIND Toolkit.

The trade-off of sparse sampling of the wind rose is that the AEP from numerical integration is highly sensitive to the number
of bins chosen. The AEP varies by as much as 40% as we reduce the number of wind direction bins from 72 to 9. To reduce





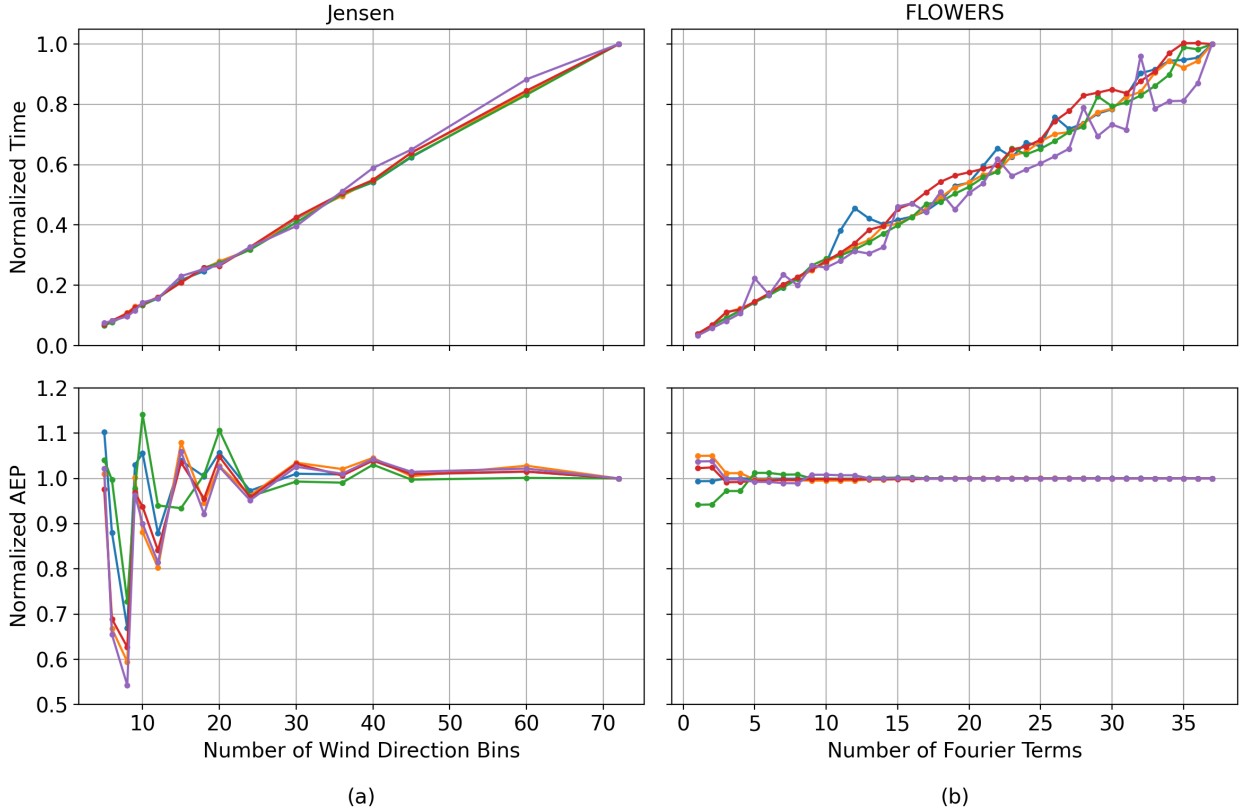

**Figure 5.** The effect of the resolution of wind direction bins in the Jensen model (left) and number of Fourier modes in FLOWERS (right) on cost (top) and accuracy (bottom) of the AEP calculations. Each colored line represents one of five sampled wind roses. AEP is computed for a 7-by-3 grid of turbines with a spacing of $5D$.

the computational cost by a factor of 2, AEP fluctuates by about 2%; to reduce the cost by a factor of 5, AEP changes up to 10%. The sensitivity manifests as both overestimates and underestimates of AEP, so it is not possible to assume a conservative underestimate of AEP, for example.

The equivalent idea in the FLOWERS formulation is to reduce the number of Fourier series modes. Each term in the discrete Fourier series is a single arithmetic expression, so the cost should also scale linearly with the number of terms. Figure 5b

illustrates that the computational time for FLOWERS is proportional to the number of Fourier modes included in the solution. The lowest-frequency modes are used. In contrast to the numerical integration method, the AEP for FLOWERS is less sensitive to the number of Fourier modes. For all five wind roses, AEP remains virtually unchanged when using half of the maximum number of terms, and within 1% when using only five terms. The Fourier transform is well-suited to approximate the wind speed and frequency data and is a common tool to represent complex signals in a compact format. Reducing the number of

wind direction bins is less successful because the decrease in resolution is indiscriminate: data is aggregated in uniform bins





**Table 1.** Influence of number of Fourier modes in the FLOWERS solution for the 40 randomized AEP test cases.

| FLOWERS Fourier Terms | $N = 37$ | $N = 5$ |
|---|---|---|
| AEP Mean Difference | 7.97% | 8.00% |
| AEP Std. Difference | 6.95% | 6.96% |
| Mean Computation Time Ratio | 22.4 | 142.1 |

and averaged without considering the important features of the signal. Another useful feature of the Fourier expansion is that reducing the number of modes does not change the smoothness of the superimposed signal. A single Fourier mode is still a continuous sinusoidal function. On the other hand, reducing the number of wind direction bins directly causes a more discrete numerical integral. The more coarse the wind direction, speed, and frequency data is, the more sensitive the AEP calculation

becomes. This has significant implications for the quality and robustness of layout optimization solutions.

With better understanding of the low sensitivity of AEP accuracy to the number of Fourier terms in FLOWERS, we have the opportunity to further reduce the computational cost. To hone in on the appropriate number of Fourier terms to use, we return to the 40 randomized test cases from Section 3.2. Table 1 compares the percent difference in AEP and the ratio of computation time between Jensen and FLOWERS with the maximum number of Fourier terms ($N = 37$) and a truncated Fourier series

($N = 5$). The mean and standard deviation of the AEP difference between FLOWERS and Jensen remain virtually unchanged. Only the FLOWERS AEP is computed differently between these two cases, which indicates that the FLOWERS AEP is insensitive to the number of Fourier modes included in the solution. The advantage is that the FLOWERS solution with five Fourier terms is almost 150 times faster to compute than the numerically-integrated Jensen method.

We therefore recommend using this truncated FLOWERS solution. By using only one-eighth of the Fourier terms, there is a

reduction in cost of roughly a factor of 6 with virtually no trade-off in accuracy. There is no reason to use the extended Fourier series if it only increases the computational cost of the FLOWERS solution.

## 4 Optimization Comparison

Consider 9 turbines placed within a square boundary of side length $12D$. The wind is coming from the left, with a fixed speed of 8 m/s. We compare two optimizations with different objective functions: one with the FLOWERS AEP, and the other with

the conventional AEP via numerical integration. All other inputs and parameters in the optimization are identical: initial layout of the wind plant, wind direction, speed, and frequency distribution, wake expansion coefficient, and convergence threshold. Figure 6 shows this wind rose with three different resolutions: $1°$, $5°$, and $40°$ wind direction bins. These different resolutions are used in the following studies.

The AEP that drives the gradient-based optimization is different between both optimizers. However, we wish to compare the

quality of the optimal solutions for both without confounding the differences in AEP discussed in Section 3. In this section, we use the numerical integration method to compute the AEP produced by the initial and final layouts of both optimizers for a



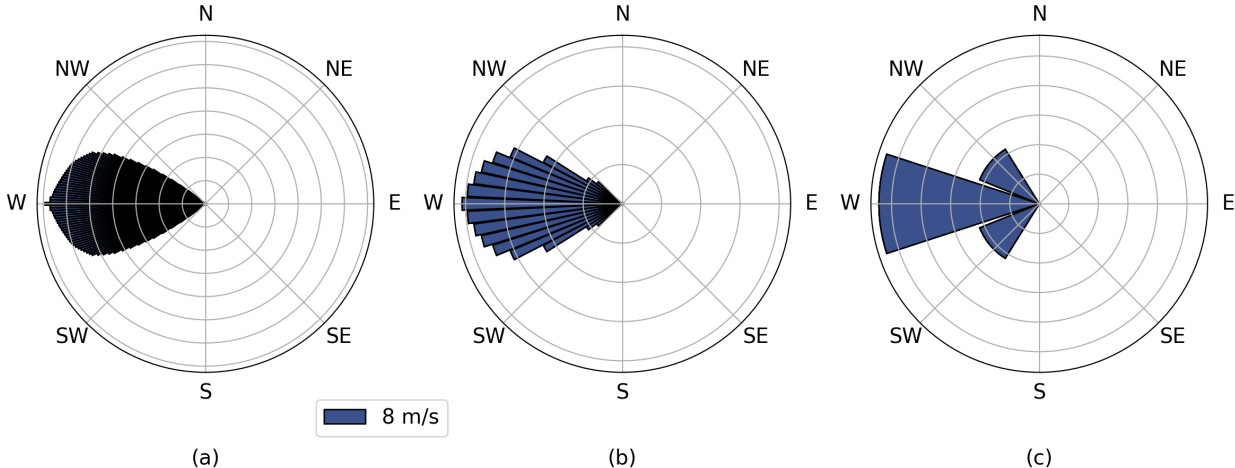

**Figure 6.** The wind rose used for the layout optimization studies performed in Section 4. The resolution of the wind direction bins varies from $1°$ (left) to $5°$ (center) to $40°$ (right). The wind speed is a constant 8 m/s.

straightforward comparison. The objective function of the FLOWERS optimizer still uses the FLOWERS AEP formulated in Section 2.2, but the AEP that we are reporting here is computed in the same fashion as the Jensen AEP.

### 4.1 FLOWERS and Jensen

The first comparison is against the integrated Jensen model. $5°$ wind direction bins (i.e. $B = 72$) are used for the Jensen model (see Figure 6b) and $N = 5$ Fourier terms are used for FLOWERS.

Figure 7 shows the results of the optimization for a randomized initial layout of the wind plant. The pattern of turbine placement is qualitatively different between the FLOWERS and Jensen optimizers. The Jensen optimization focuses on maximizing the streamwise spacing of the turbines by placing all turbines on either the leading or lagging edge of the wind farm's boundary.

One reason why the Jensen optimizer favors this type of solution is that wake-added turbulence is included in the modeling framework, so maximizing the streamwise spacing of the turbines improves the wake recovery for downstream turbines. On the other hand, the FLOWERS solution focuses on the spanwise spacing of the turbines, placing them such that eight turbines are unaligned with respect to the predominant wind direction from the left. As expected, the FLOWERS optimization is performed faster than Jensen by about a time factor of 18 (814 s versus 46.0 s). The FLOWERS solution actually achieves a more optimal

AEP gain of 15.0% versus 12.9% for Jensen.

To investigate this result more generally, we consider nine additional multistart cases with randomized initial conditions (ten in total). Figure 8 displays the computation time and AEP gain for these ten cases. The previous example from Figure 7 corresponds to Case 3 here. On average, FLOWERS is about 59 times faster than the Jensen optimizer. Also, FLOWERS achieves an AEP gain that is on average 1% higher than Jensen.





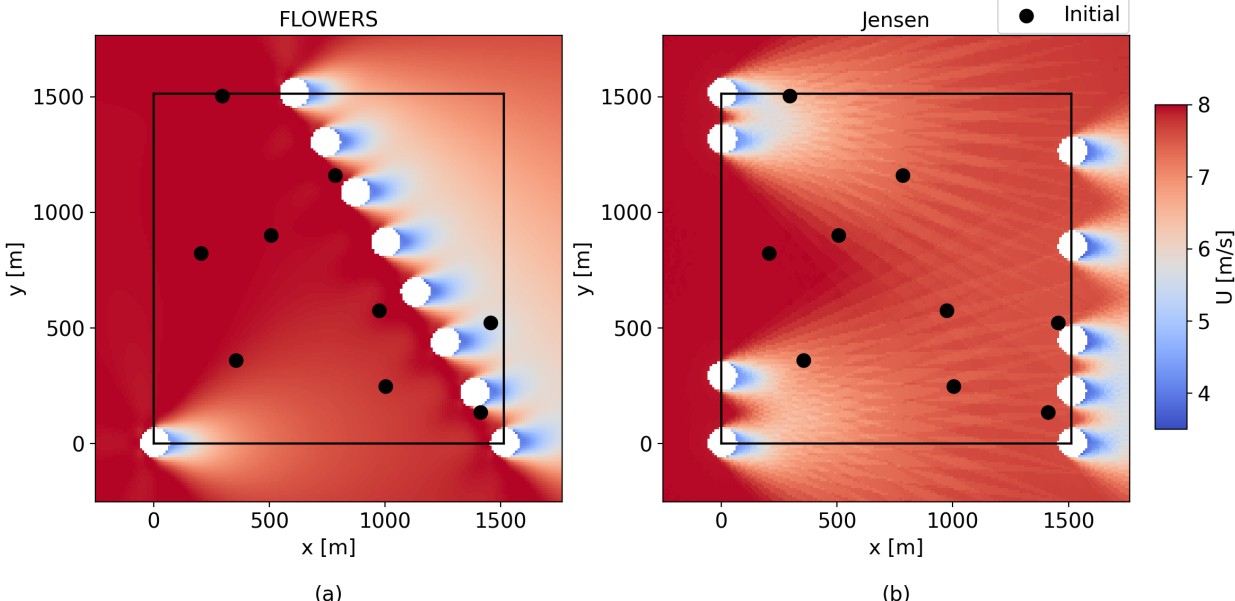

**Figure 7.** The optimal layouts for the FLOWERS and Jensen optimizers ($B = 72$). The FLOWERS solution required 46.0 s and achieved an AEP gain of 15.0%. The Jensen solution required 814 s and achieved 12.9% AEP gain. The black dots denote the initial layout.

The superior performance of FLOWERS compared to the Jensen optimizer connects back to the smooth nature of the formulation. The FLOWERS optimization space is smooth and continuous because of the Fourier transform and analytical integration. On the other hand, the Jensen optimization space is coarse because of the discrete model and numerical integration. The gradient-based optimizer thrives in the smoother optimization space of FLOWERS. More refined adjustments of the turbine positions are possible and the optimizer is less likely to become stuck in local optimal solutions in the smooth landscape. In the

discrete space of the Jensen optimization, it is more difficult for the optimizer to explore the optimization space with equivalent precision and efficiency.

To test the effect of wind rose resolution on the optimization performance, we use $B = 360$ wind direction bins for the Jensen model, as seen in Figure 6a; we maintain $N = 5$ Fourier terms for FLOWERS. Figure 9 displays the computation time and AEP gain for these ten new cases. FLOWERS is now about 200 times faster than the Jensen optimization on average; the

relative improvement in computation time is due to the Jensen AEP calculation covering five times more wind direction bins. The average AEP gain in FLOWERS is about 1.8% higher than that for Jensen. The particular AEP gains that are achieved by each optimizer in this limited sample size of 10 cases are sensitive to the initial layout of the wind farm. Regardless, the overall result is that there is a negligible change in the quality of the solution that the Jensen optimizer achieves with a higher-resolution wind rose.

We should note that the 10 randomized initial layouts tested here represent a limited sample size for a multistart study. The results still allow for a comprehensive discussion of the differences between the FLOWERS and Jensen models. Future

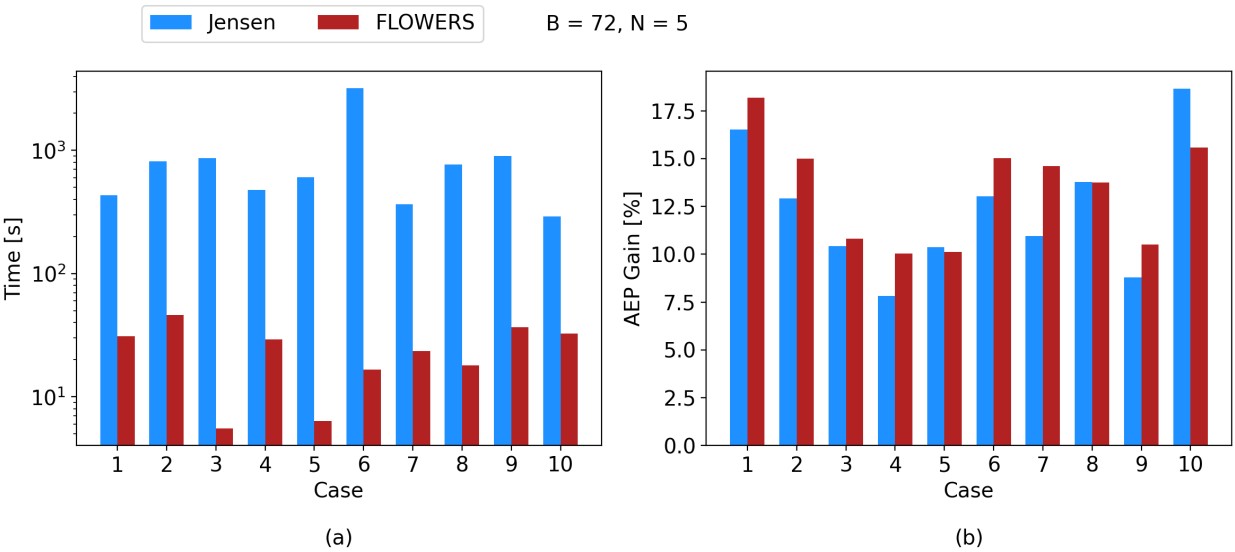

**Figure 8.** A comparison of cost and performance for 10 cases with randomized initial conditions. Five Fourier modes are used for the FLOWERS solution and 72 wind direction bins are used for the Jensen model integration. FLOWERS is on average about 59 times faster than Jensen and achieves an AEP gain that is 1% higher.

work could expand on the scope of the multistart experiments to investigate whether FLOWERS converges to more consistent solutions than Jensen, and also whether FLOWERS converges to a solution in fewer iterations than Jensen.

### 4.2 FLOWERS and Gauss

Comparing FLOWERS to a smoother wake model, in this case the Gaussian wake model (Bastankhah and Porté-Agel, 2014):

$$\frac{U(x,y)}{U_\infty} = \left(1 - \sqrt{1 - \frac{C_T}{8(kx + \epsilon)^2}}\right) \exp\left(-\frac{y^2}{2(kx + \epsilon)^2}\right), \tag{13}$$

further highlights the characteristics of the FLOWERS method relative to the conventional numerical integration approach. Instead of discrete boundaries of the wake in the tophat model, the Gaussian model is smooth in space. However, as a trade-off for the improved detail of the Gaussian profile, the cost of a function evaluation is higher. We use a sum-of-squares wake deficit
superposition for the Gaussian wakes. Every other parameter of the optimization study is the same as the previous experiment, including 1° wind direction bins (i.e. $B = 360$).

The results of one of the optimization studies is shown in Figure 10. The Gauss optimal solution is smooth, without the discrete boundaries of the Jensen model, and the layouts are qualitatively more similar to FLOWERS, with most turbines placed in angled rows. The similarity of these solutions suggests that the optimization spaces are similar despite using different
wake models and AEP calculations. The similarity is not merely qualitative: the AEP gain for FLOWERS is 7.9% and Gauss





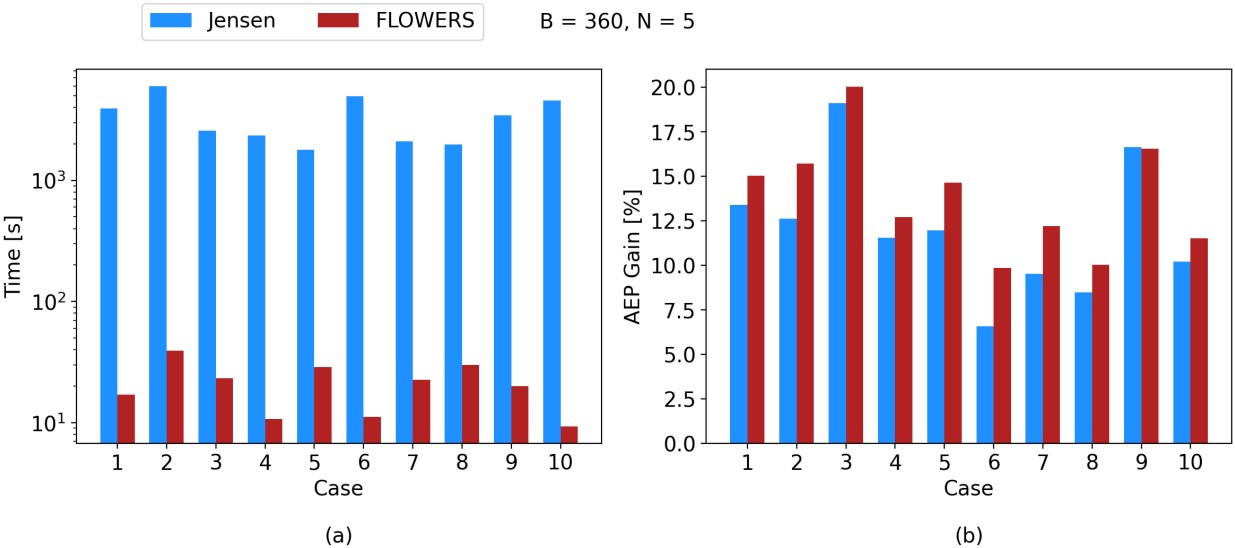

(a)  (b)

**Figure 9.** A multistart study, now for the Jensen model with $B = 360$ wind direction bins. The Jensen optimization now takes about 203 times longer than FLOWERS and achieves an AEP gain that is 1.8% lower.

is about 7.3%. The fact that the Gaussian gain is 0.6% lower than that for FLOWERS is not necessarily a sign that the Gauss optimization is subpar, but rather a byproduct of the many local optima. There are many possible layouts that satisfy the spacing constraints and achieve similar AEP performance, so the solutions given here likely represent local optima, not a global optimum. Since the two optimizations evaluate different objective functions and operate in different solution landscapes,

they cannot be expected to arrive at identical solutions. However, the trade-off for improved performance by using the Gauss wake model with numerical integration is in computational cost. The Gaussian optimization took 16,700 s (4.6 hr), while the FLOWERS optimization only required 46.4 s. This is an improvement by a factor of 360 for FLOWERS.

Figure 11 displays the results for the 10 multistart cases. Figure 10 is Case 1 in these plots. On average, the Gaussian optimization takes 690 times longer than FLOWERS. The average AEP gain in FLOWERS is 0.1% higher than Gauss. The

takeaway from this small difference in AEP gain is that FLOWERS and Gauss produce comparable solutions, not that FLOWERS is better at finding a more optimal solution. A stricter convergence criterion would force the optimizers to search longer for the global solution, which would likely cause these AEP gains to grow even more similar.

This experiment suggests that the smoothness of the Gaussian model compared with the Jensen model is the most likely explanation for the improved performance. While the number of wind direction bins is unchanged, the flow field for each

simulation is smoother with the Gaussian model. When a turbine's position is adjusted, there is no binary switch between being within the wake or outside of it; this discrete change in wake velocity would cause the AEP to be sensitive to slight perturbations in the turbine positions. On the other hand, in the Gaussian model, a small change in turbine position results in





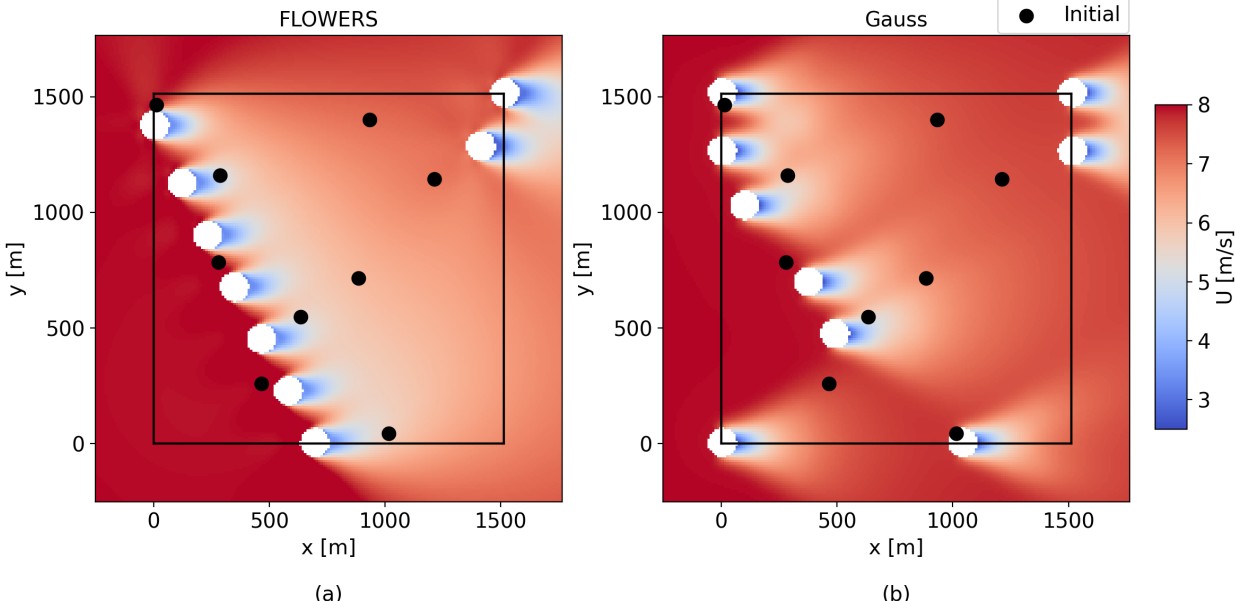

**Figure 10.** The optimal layouts for the FLOWERS and Gauss optimizers (with $B = 360$ wind direction bins). The FLOWERS solution required 46.4 s and achieved an AEP gain of 7.9%. The Gauss solution required 16,700 s and achieved 7.3% AEP gain. The black dots denote the initial layout.

a similarly small change in the wake deficit because of the smooth profile. This continuity produces a more smooth solution space, which enables the optimizer to move along more subtle gradients and achieve more optimal solutions than the Jensen
optimizer.

While the quality of solutions between FLOWERS and the Gauss optimizer are comparable, there is no contest in terms of cost. The FLOWERS optimization is almost three orders of magnitude faster than the Gauss optimizer and can produce optimal solutions with equivalent performance. Moreover, we are only comparing the results for a wind plant with nine turbines. As illustrated in Figure 3a, the computational cost scales with the number of turbines more sharply with the numerical integration
approach. It is possible that this factor of 690 could grow to a factor of 1,000 or more for a larger wind farm.

We have demonstrated that the FLOWERS AEP is insensitive to the number of Fourier series terms, and have used the truncated series to achieve similar performance to the Gauss optimization. We also previously showed that the AEP calculated from numerical integration is extremely sensitive to the resolution of the wind direction bins. For a fair and comprehensive comparison, the Gauss optimization should be performed with a limited number of wind direction bins to mimic the reduction
in cost that was implemented in FLOWERS. To match $N = 5$ Fourier terms, $B = 9$ wind direction bins are now used in the Gaussian optimization, as seen in Figure 6c. The results in this case are shown in Figure 12. The reduction in wind rose resolution brings the cost of the Gaussian optimization down significantly such that the FLOWERS optimizer is only about 6 times faster on average. However, the AEP gain of the optimal solutions for this Gaussian optimizer with a coarse wind rose





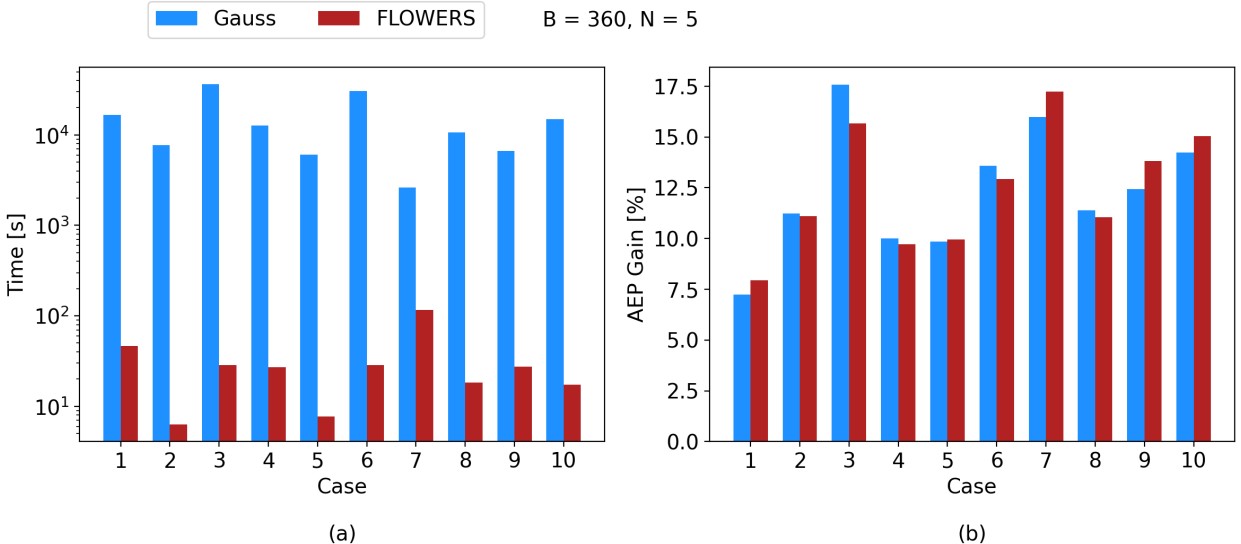

(a)  (b)

**Figure 11.** Multistart experiment for FLOWERS against the Gauss model with $B = 360$ wind direction bins. FLOWERS is about 690 times faster than Gauss on average, and the AEP gain between the two optimizers is within 0.1% on average.

is poor. The AEP gain for FLOWERS is 7% higher than Gauss on average, which is an improvement of a factor of 2.5. This
experiment proves that the Gaussian optimizer cannot achieve greater computational efficiency by manipulating the resolution
of the wind rose without substantially impacting the quality of its solutions.

## 5  Potential Model Improvements

As we have demonstrated, FLOWERS is able to match the performance of conventional layout optimization methods despite
simplifications in its formulation. The discrepancies in AEP estimates between FLOWERS and the integration of the Jensen
and Gaussian models did not inhibit its application to the optimization problem. However, we could enhance the accuracy of
FLOWERS by improving the following:

- **Power integral:** We introduced a simplification to make the integration of turbine power tractable by computing AEP
  as a function of $\overline{U}^3$ rather than $\overline{U^3}$. We would reexamine the integration of Eq. 11 to avoid this simplification, which
  introduces errors when wind speed varies across different wind directions.

- **Coefficient of power:** $C_p$ is currently defined as a function of the average wake velocity, making it a constant. We aim to
  incorporate $C_p$ as a function of the wake velocity for each wind direction such that average power is computed exactly:
  $\overline{P(U)} = \frac{1}{2}\rho A \overline{C_p(U)U^3}$.

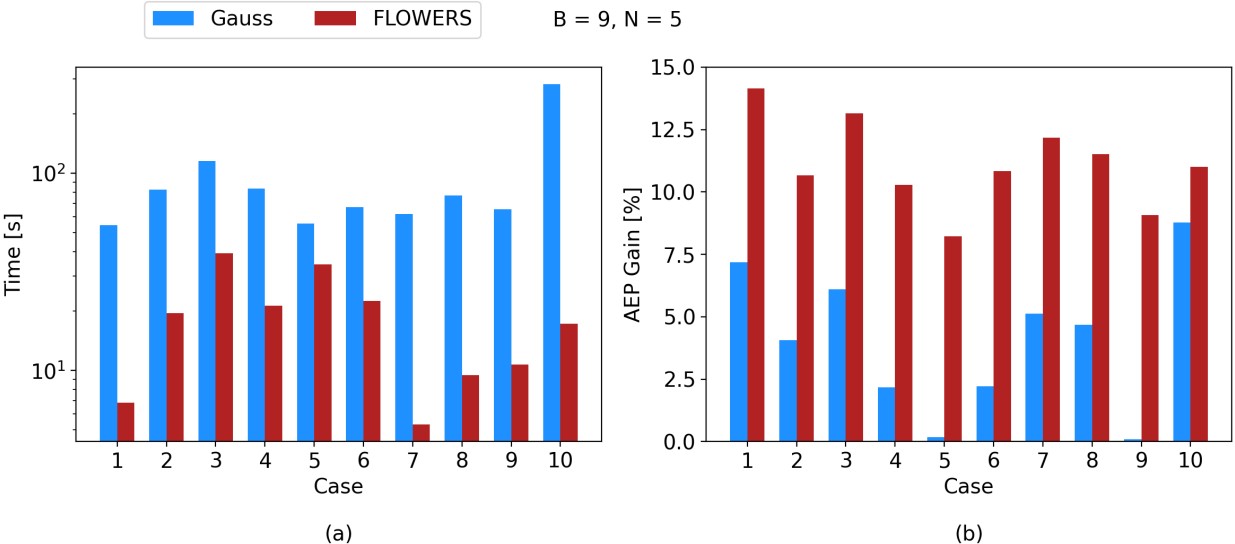

**Figure 12.** Ten randomized cases for FLOWERS against the Gauss model with $B = 9$ wind direction bins. FLOWERS is only about 6 times faster than Gauss now, on average. The average AEP gain for FLOWERS is about 11%, but only about 4% for Gauss.

- **Local flow conditions:** We can define the wake velocity deficit relative to the local flow velocity rather than the freestream, which will better capture the influence of upstream turbines and development of the flow as it moves through the wind farm. This improvement is particularly expected to improve results in aligned cases such as the one discussed in Section 3.1.

- **Gaussian model:** We currently integrate a classical tophat wake deficit model. We expect that the AEP estimates would be more accurate by integrating a Gaussian wake model instead.

## 6   Conclusions

The objective of this paper was to develop a novel analytical formulation of annually-averaged wake velocity to use in a layout optimization problem and demonstrate its effect on reducing the computational cost of these studies. We derived the equations for the analytical integration of the tophat wake deficit model. The wind speed and wind direction frequency distributions were expressed as a Fourier series to facilitate the integration.

The annually-averaged wake velocity was used to compute AEP. We approximated the average power by using the average wake speed cubed rather than an average of the cube of the wake speed, which introduces error when there are pronounced wakes or the wind speed varies significantly across different wake directions. Also, the local wind speed's effect on power





production was not accounted for, and a constant power coefficient was assumed. These simplifications introduced error that led to the AEP computed in FLORIS differing from the numerical integration approach by about 8%.

Fortunately, these limitations in the accuracy of the FLOWERS AEP do not preclude its use in the optimization problem.
The FLOWERS optimizer built around the Jensen wake model finds optimal wind plant layouts with AEP comparable to an optimizer that numerically integrates a Gaussian wake model. This finding is unexpected but promising because it implies that the mathematical formulation behind FLOWERS compensates for a more simplistic wake model to achieve similar results as a more sophisticated wake model. The clear advantage of FLOWERS, then, is the robust layout optimization performance while achieving a reduction in computational cost of over two orders of magnitude. We believe that this improvement in computation
time will scale better with wind farms containing more than the nine turbines studied here.

This achievement could translate to the difference between running an optimization study in 10 minutes versus 5 days, or between running the study on a personal laptop versus a high-performance computer cluster. This technique could open the door for other areas of research in layout optimization, including optimization under uncertainty, by making these studies more accessible and less costly. Moreover, the new conceptualization of the wake velocity deficit could inspire brand new areas of
research in wake modeling and wind plant control and optimization.

This paper serves as a foundation for future work on the FLOWERS formulation. Since the motivation of this approach was to improve computational cost, one avenue to explore is further optimization of the FLOWERS code. Wind plant layout and yaw steering co-design is a popular area of research, and another potential application for FLOWERS if yaw deflection models could be included in the formulation. We also plan to validate the performance of the FLOWERS optimization against
high-fidelity simulations.

*Author contributions.* **M.J. LoCascio:** Software, Investigation, Writing - Original Draft **C.J. Bay:** Conceptualization, Methodology, Software **M. Bastankhah:** Methodology, Formal Analysis, Writing - Review & Editing **G.E. Barter:** Funding Acquisition, Writing - Review & Editing **P. Fleming:** Funding Acquisition **L.A. Martínez Tossas:** Conceptualization, Methodology, Supervision.

*Competing interests.* The authors declare that they have no conflict of interest.

*Acknowledgements.* The authors would like to thank Bart Doekemeijer, Nicholas Hamilton, Jennifer King, Patrick Moriarty, Rafael Mudafort, Eric Simley, and P.J. Stanley for their feedback and support. MB acknowledges funding from Innovate UK (grant no. 89640). A portion of the research was performed using computational resources sponsored by the U.S. Department of Energy's Office of Energy Efficiency and Renewable Energy and located at the National Renewable Energy Laboratory. This work was authored in part by the National Renewable Energy Laboratory, operated by Alliance for Sustainable Energy, LLC, for the U.S. Department of Energy (DOE) under Contract No. DE-
AC36-08GO28308. Funding provided by the U.S. Department of Energy Office of Energy Efficiency and Renewable Energy Wind Energy Technologies Office and the U.S. Department of Energy Office of Science Office of Workforce Development for Teachers and Scientists



under the Science Undergraduate Laboratory Internships Program. The views expressed in the article do not necessarily represent the views of the DOE or the U.S. Government. The U.S. Government retains and the publisher, by accepting the article for publication, acknowledges that the U.S. Government retains a nonexclusive, paid-up, irrevocable, worldwide license to publish or reproduce the published form of this work, or allow others to do so, for U.S. Government purposes.




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
