# Peer review of "FLOW Estimation and Rose Superposition (FLOWERS): An integral approach to engineering wake models"

_Wind Energy Science, 2021_

## Referee Comment (RC2)

**Review of *FLOWERS: An integral approach to engineering wake models* by Michael J. LoCascio et al.**

Reviewer: M. Paul van der Laan, DTU Wind Energy

December 2, 2021

The authors propose to apply an analytical integration method for evaluating the annual energy production (AEP) of a wind farm and compare its performance against the more common numerical integration method.

The article is well written and contains novel ideas. However, I do have some major concerns with the assumptions of proposed method and the employed methodology for comparing the analytical and the traditional AEP integration methods. More detailed comments are listed below; they need to be addressed before the article can be considered for publication in Wind Energy Science.

**Main comments**

1. My main concern with this work is that one needs to use constant thrust and power coefficients (for all wind speeds) in order to be able to perform an analytical integration of the AEP by integrating the velocity deficits for all wind directions and wind speeds. Wind turbines do not have constant thrust and power coefficients above rated wind speed, which makes the proposed method difficult to use for wind farm layout design and optimization. The authors propose to include more realistic thrust and power coefficients in future work, but I not see how this can be done, since the current derivation of the analytical model only holds for constant thrust and power coefficients. In addition, the errors from using constant thrust and power coefficients for a wind farm layout optimization are not properly addressed because you only investigate a single below rated wind speed where the assumption is not violated significantly. Therefore, you need to add a realistic case as well (including all wind speeds, for example from 4-25 m/s) to access the errors associated to the constant thrust and power coefficients in terms of AEP.

2. The motivation of this work is to speed up an AEP calculation using an engineering wake model. The authors report a calculation time of 16700 s for a wind farm layout optimization using a numerical AEP integration within FLORIS using a wind farm of 9 wind turbines, 360 wind directions and a single wind speed. It is not clear how many different wind farm layouts are evaluated within the optimization process, but even if this was 1000 then the calculation time per layout seems to be quite long considering the small number of turbines and the use of only a single wind speed. In PyWake [2], a single AEP calculation for a larger wind farm (e.g. 80 turbines) using 360 wind directions and 22 wind speeds typically takes less than a second, so one could easily perform a wind farm layout optimization using a numerical integral. I suspect that the numerical implementation in FLORIS could be improved significantly. PyWake is an open source tool, so you could test its AEP calculation speed yourself (the script below took 0.64 s to calculate the AEP of the Horns Rev I wind farm on DTU's Sophia cluster using a single CPU):

```
import time
import numpy as np
from py_wake.examples.data.hornsrev1 import Hornsrev1Site
```

```
from py_wake import NOJ
from py_wake.examples.data.hornsrev1 import wt_x, wt_y, HornsrevV80, Hornsrev1Site
wt = HornsrevV80()
site = Hornsrev1Site()
wf_model = NOJ(site, wt)
**360 wind directions**
wd = np.arange(0.0, 360.0, 1.0)
**22 wind speeds**
ws = np.arange(4.0, 26.0, 1.0)
starttime = time.time()
**Calculate AEP**
sim_res = wf_model(wt_x, wt_y, wd=wd, ws=ws)
aep = float(sim_res.aep(with_wake_loss=True).sum()
endtime = time.time()
print('AEP:', aep)
print('Total time:', endtime - starttime, 'sec')
```

3. The authors mention that the analytical integration method is more smooth and therefore better suited for optimization. While this is indeed true, one could also consider a numerical integration method using the Gaussian wake model including analytical functions for all the discrete input (Weibull, wind rose, power and thrust coefficient curve), see for example [1]. (This reference also investigates how many flow cases are necessary for an AEP calculation, which you also look at, so you could cite it.)

4. Figure 5: It is interesting to see that the analytical integration works well for only a small number of Fourier modes.

5. Line 72: You definition of the wake region,

$$W(x, y) = \begin{cases} 1, & \text{if } |y| \leq kx + 1 \\ 0, & \text{otherwise} \end{cases} \qquad (1)$$

does not hold because it would imply that there is a wake region upstream (for negative values of $x$). So you would need to be more precise, for example:

$$W(x, y) = \begin{cases} 1, & \text{if } |y| \leq kx + 1 \text{ and } x \geq 0 \\ 0, & \text{otherwise} \end{cases} \qquad (2)$$

which applies to a wind direction of $270°$.

6. Equation (1): You forgot to define the origin, I guess it is the wind turbine location. In addition, the original NOJ model use $1 + 2k$ so your wake expansion coefficient is twice as large as the standard model. I think you should mention this to avoid confusion.

7. Sections 3.1 and 3.2: A difference of 1.8% and 8% in AEP is quite a lot considering the amount of money 1% AEP represents in a large wind arm. Therefore, a wind farm developer would go for the more precise numerical method instead of the analytical method if the error is that large.

8. Section 4.1: When you compare the results of optimized AEP between FLOWERS and Jensen, do you post evaluate the AEP of the optimized layout from FLOWERS using the numerical approach? If not, then you cannot really compare the AEP values. In addition, why do you use wake added turbulence for the Jensen model and not for the FLOWERS model? Wouldn't be more fair to apply the Jensen model in the same way (so without added wake turbulence)?

9. Can you use your analytical AEP integration model if the wake superposition method is non-linear? The wake superposition model is important part of AEP calculation using engineering

wake models and one might typically want to aplly several wake superposition methods in order to get an estimate of the model uncertainty.

**Minor comments**

1. The title could be more clear. When I first read it I did not think of an analytic AEP integration method.

**References**

[1] Murcia, J. P., Réthoré, P. E., Natarajan, A., and Sørensen, J. D. How many model evaluations are required to predict the AEP of a wind power plant? *Journal of Physics: Conference Series*, 625:012030, jun 2015.

[2] Pedersen, M. M., van der Laan, P., Friis-Møller, M., Rinker, J., and Réthoré., P. DTUWindEnergy/PyWake: PyWake, 2019.

---

## Author Comment (AC1)

**FLOWERS Reviewer Comments**

Michael J. LoCascio et al.

February 19, 2022

**CC1**

Thank you for a very interesting paper. I really like the idea of fitting the wind rose with a fourier series to avoid the wake rays which is clearly seen when applying the tophat model for a limited number of wind directions. I have a few comments, questions and corrections:

We appreciate the positive feedback from the reviewer. The responses are below and marked in blue.

1. Is it possible to include local flow conditions without a time-consuming iterative approach?

   That's a good question, and we have discussed using an iterative approach as part of future work. We have not included local flow conditions in this work, so we have not needed to implement an iterative approach to solve for the velocity deficits throughout the wind farm. We have adjusted the text in the manuscript to:

   "Local flow conditions: We can define the wake velocity deficit relative to the local flow velocity rather than the freestream, which will better capture the influence of upstream turbines and development of the flow as it moves through the wind farm. This improvement will require an iterative approach and is particularly expected to improve results in aligned cases such as the one discussed in Section 3.1."

2. As I understand it, you will try to fix the power integral and allow a wind-speed dependent cp. Will this solution capture the fact that an annual wind distribution normally also includes wind speeds above rated with lower cp and is it possible to also allow a wind-speed dependent ct?

   Yes, it is possible to allow for a wind-speed dependent $C_T$ and $C_p$. Since we utilize a lookup table for these coefficients based on wind speed, we would account for wind speed domains above the cutoff speed. In our formulation, $C_T$ can be made dependent on the freestream wind speed by including it in the Fourier expansion as a function of wind direction. We have changed the derivation to include this result:

   "We define $g(\theta') = \frac{1}{2\pi} U_\infty(\theta') f(\theta')$ and $h(\theta') = \frac{1}{2\pi}\left[1 - \sqrt{1 - C_T(U_\infty(\theta'))}\right] U_\infty(\theta') f(\theta')$. If we expand $g(\theta')$ and $h(\theta')$ with Fourier series:

   $$g(\theta') = \frac{c_0}{2} + \sum_{n=1}^{N} c_n \cos(n\theta') + d_n \sin(n\theta')$$

   $$h(\theta') = \frac{a_0}{2} + \sum_{n=1}^{N} a_n \cos(n\theta') + b_n \sin(n\theta'),$$

   then the wind rose is defined continuously rather than discretely. The Fourier coefficients $a_0$, $a_n$, $b_n$, $c_0$, $c_n$, and $d_n$ can be easily found for a given $g(\theta')$, $h(\theta')$, and $N$."

   Then, the solution for $\overline{U(r,\theta)}$ is computed using the Fourier expansion of these two separate functions:

$$\overline{U(r,\theta)} = \int_{-\pi}^{\pi} g(\theta')\, d\theta' \quad - \sum_i \int_{-\theta_c}^{\theta_c} h(\theta_i - u) \left[ \frac{1}{(kr+1)^2} + \frac{kru^2}{(kr+1)^3} \right] du.$$

The difference between this approach and the conventional method is also clarified in the manuscript:

"In FLOWERS, we neglect local wind conditions when computing the wake velocity and instead normalize the velocity deficit by the global freestream velocity and assume a uniform $C_p$ and $C_T$ across all turbines; the Jensen numerical integration implementation, on the other hand, normalizes the velocity deficit with and computes $C_p$ and $C_T$ as a function of the local wind velocity."

In our formulation, $C_p$ is treated as a function of the average wake speed $\overline{U}$. We can use the wake speed instead of the freestream speed here because we have already calculated $\overline{U}$ by the time we estimate the power production of the turbine, and we expect this average wake speed to be a better approximation to look up $C_T$ than the freestream wind speed. Allowing $C_T$ and $C_p$ to be a function of wind speed, even if the use of the freestream wind speed is less accurate, would enable the model to reflect the effect of wind speed magnitude across the wind farm (i.e. uniformly high wind speeds over the cutoff speed or low wind speeds near the cut-in speed). The model derivation and results have been adjusted to reflect these changes. However, we cannot treat $C_T$ and $C_p$ as a function of the local wind speed in our current formulation. We plan on exploring this idea in future work.

3. I assume the codes are implemented in python using numpy. The problem is that a python loop is so much slower than a numpy vector operation. The time comparison is therefore only fair if the none of the codes contain extra python loops compared to the other, which could be vectorized.

This is a good point, and we plan on comparing to a vectorized code in future work. The objective of this paper was to derive the model and illustrate the potential for computational efficiency improvements over another conventional platform. We do not intend for this work to be a comprehensive comparison between softwares, or to underscore deficiencies in common implementations of AEP calculations. We have added this note to the manuscript:

"We should note that the implementation of the Jensen numerical integration in FLORIS Version 2 is non-vectorized (i.e. calculations are mostly performed through for-loops instead of vector operations). Comparisons in computation time with a vectorized code such as PyWake would likely show a smaller discrepancy. However, the code implementation of FLOWERS is not fully vectorized either, so it is a fair comparison of FLOWERS to the Jensen numerical integration to showcase the performance of our newly-developed model."

4. Line 48: 180 > 270.

Thank you for catching that. The correction has been made in the manuscript:

"For example, in Figure 1b the frequency of the 270° wind direction is greater than that for 225° (and the freestream wind speed is held constant), and so the velocity contour plot shows the wake directed horizontally with a stronger velocity deficit compared with the angled wake."

5. Section 3.1: You indicate that you are using local waked wind speed instead of free-stream wind speed when scaling the Jensen deficit, is that correct?

That is correct, and has been clarified in the manuscript:

"...the Jensen numerical integration implementation, on the other hand, normalizes the velocity deficit with and computes $C_p$ and $C_T$ as a function of the local wind velocity."

6. Line 177: "FLOWERS is only computing the velocity through each turbine at a single point instead of an array of points on the rotor area." How many points are evaluated with the Jensen model?

There are 25 points per rotor. The following text has been added to the manuscript:

"This difference scales with the size of the wind farm, in part because FLOWERS is only comput-ing the velocity through each turbine at a single point instead of an array of points on the rotor area (25 points per rotor in the conventional wake model)"

7. Line 209: speed > direction?

We meant wind speed and frequency for each wind direction. This wording has been clarified in the manuscript:

"The Fourier transform is well-suited to approximate the wind speed and frequency data for each discrete wind direction and is a common tool to represent complex signals in a compact format."

8. Line 245: "One reason why the Jensen optimizer favors this type of solution is that wake-added turbulence is included in the modeling framework, so maximizing the streamwise spacing of the turbines improves the wake recovery for downstream turbines." You claim to have identical wake expansion factor, which I assume means a constant factor. This conflicts with improved wake recovery due to wake-added turbulence. As far as I can see, the Jensen simulation gives a interference mesh of wake "fingers", which results in many local optima, where the optimizer gets stuck as you also write in section 4.1

This is a good point. What was meant is a nominal wake expansion rate based on the ambient tur-bulence intensity, which is identical to the wake expansion rate for freestream, unwaked turbines in the Jensen model. The inclusion of wake-added turbulence leads to an increased wake recovery rate in waked turbines within the Jensen model. This language has been clarified in the text:

"The AEP for the numerical integration approach is computed using the Jensen wake deficit model with the same nominal (i.e. based on ambient turbulence intensity and excluding wake-added turbulence) wake expansion coefficient as in FLOWERS ($k = 0.05$)."

9. Line 253: "corresponds to Case 3". It looks more as Case 2.

Good catch on a typo. Thank you for the attention to detail. Since refreshing the results based on this feedback, we have changed the case we use as a representative example to Case 4.

10. Is it correct understod that all 4x10 runs in figure 8, 9, 11 and 12 would end up with the same AEP gain (same optimized layout) if the optimizations were perfect? Maybe the same 10 cases could be compared in the one figure for the five different approaches (Jensen_72, Jensen_360, FLOWERS, Gauss_360, Gauss_9) if it does not spoil your nice flow of the story.

Yes, one could argue that there is a global maximum in AEP and with sufficiently many iterations, the optimizer could consistently obtain a globally optimal solution. However, with our simplified test case here with a single predominant wind direction, there are many wind farm layouts that achieve similar degrees of wake avoidance. These layouts are local optima and are output as solutions given our finite optimization convergence tolerance. This simple wind rose and wind farm boundary is used to illustrate the new model formulation in an easily-digestible format. We would expect fewer local optima in an optimization problem with a more realistic wind rose where it is harder to achieve wake avoidance.

The initial conditions across the 40 multistart cases (4x10) are all randomized–we do not use the same 10 random starts across the four different optimization setups (Jensen 72, Jensen 360, Gauss 360, Gauss 9). So, Case 1 for each approach has a different initial AEP. Since AEP gain is defined as percentage gain relative to the initial value (a definition has been added to the manuscript for clarity), then the AEP gain will naturally vary among every case. Only by direct comparison against the same initial condition, as is done between FLOWERS and the corresponding numerical integration approach for each set of 10 multistarts, is the AEP gain a valid metric. We add this sentence to the manuscript to reinforce this idea:

"Finding a global optimum in this unrealistic test case would require a stricter convergence crite-rion. We note that a more realistic wind rose would likely not enable so many local optima in the solution space, so this artifact should not be as pronounced in practice."

**RC1**

This is a generally well written paper and presents an apparently new and interesting method for wind farm AEP calculations. I think that some of the content could be improved and my comments are below.

We thank the reviewer for their positive comment and constructive feedback. The responses are below and marked in blue.

1. Abstract

   - What does gradient-based mean? This doesn't seem to be explained anywhere within the article.
     Definitions of gradient-based and gradient-free optimization algorithms have been added to the introduction of the manuscript:

     "Gradient-based optimization algorithms leverage the derivative of the objective function to choose search directions for optimal solutions, while gradient-free optimization only evaluates the objective function (thereby avoiding its derivatives) and is useful for discontinuous and noisy functions."

   - Perhaps consider replacing the sentence: "The analytical integral and the use of a Fourier expansion to express the wind speed and wind direction frequency create a more smooth solution space for the gradient-based optimizer to excel compared with the discrete nature of the existing weighted-averaging power calculation." with something like "The analytical integral and the use of a Fourier expansion to express the wind speed and wind direction frequency create a relatively smooth solution space for the gradient-based optimizer in comparison to the existing weighted-averaging power calculation."
     We thank the reviewer for the suggestion. The sentence has been adjusted accordingly to:

     "The analytical integral and the use of a Fourier expansion to express the wind speed and wind direction frequency create a relatively smooth solution space for the gradient-based optimizer to excel in comparison to the existing weighted-averaging power calculation."

   - It is not clear why the "weighted-averaging power calculation" is discrete.
     We clarify that the wind directions and speeds are discrete, signaling that the power calculation is also discrete. The new sentence in the abstract is as follows:

     "The conventional method to compute AEP for a wind farm is to first evaluate power production for each discrete wind direction and speed using either computational fluid dynamics simulations or engineering wake models. The AEP is then calculated by weighted-averaging (based on the wind rose at the wind farm site) the power produced across all wind directions."

2. Introduction

   - Should 'tophat' be 'top-hat'?
     The 'top-hat' nomenclature has been added to the manuscript.

   - Line 30. What are gradient-free algorithms?
     As mentioned above, we have added a definition for gradient-free algorithms to the manuscript.

   - Line 46. The term "non-zero" before wind direction seems inappropriate. Perhaps remove the term altogether or replace with 'discrete'.
     "Non-zero" has been changed to "predominant" to indicate that the frequency of all the other discrete wind directions is negligible:

     "Figure 1a shows the velocity distribution when the wind rose contains only one predominant wind direction."

3. Mathematical Formulation

- Line 69. To "the streamwise and spanwise position" I think you should add "with respect to the wind direction theta', where theta' is the wind direction in the X,Y frame"

  We clarify the definition of the Cartesian coordinate system before allowing the wind direction to vary:

  "To derive a mathematical formulation for the time-averaged flow distribution, we use the classical Jensen (top-hat) wake deficit model:

  $$\frac{U(x,y)}{U_\infty} = 1 - \frac{1 - \sqrt{1 - C_T}}{(kx + 1)^2} W(x, y),$$

  where $x$ and $y$ are the streamwise and spanwise position, respectively, normalized by the rotor radius with the origin at the turbine location. In this coordinate frame, the wind is coming from the negative $x$-direction."

- It seems that the following should be defined before equation (2): x = r cos(theta-theta'), y = r sin(theta-theta'). Then y/x = tan(theta-theta'). However, on line 73 it is stated that y/x = tan(theta) which doesn't seem correct. It would be worth putting this in a diagram for clarity, as in the attached figure.

  We have added the following text to the manuscript to be more precise:

  "We transform Eq. 2 from Cartesian to polar coordinates denoted by $r$ and $\theta$, where $x = r\cos(\theta)$ and $y = r\sin(\theta)$." However, we think the progression of the derivation is more clear to first detail the conversion to polar coordinates and then allow the wind direction $\theta'$ to vary, rather than combining this into a single step. A diagram has been added to the manuscript to aid the derivation:

[Figure]

(a)                                          (b)

Figure 1: Left: velocity contour plot of flow through a wind turbine for a single wind direction. Center: averaging effect of two wind directions. Right: the annually-averaged velocity flow field. Note that the wind roses (bottom) display the frequency of each wind direction with a constant wind speed of 8 m/s for every direction.

- Line 83. It should be clarified that the equation in this line comes directly from the equation in line 72, and represents the boundary of the wake velocity deficit. Again, this would benefit from a diagram.

  We now connect this definition back to the added figure:

"In our coordinate system, the wake geometry is defined by the line $\sin(\theta_c) = k\cos(\theta_c) + 1/r$, as shown in Figure 2b."

- Line 86 to 88. I don't think this sentence is really true. Surely the wake deficit is defined as the difference between the freestream velocity and the wake velocity.

  This definition has been corrected to be more precise:

  "The wake velocity deficit is defined as the difference between the freestream velocity and the wake velocity."

- Line 92 onwards at bottom of page 4. Does U_infinity(theta') imply there is only one inflow wind velocity for each direction? Where does the Weibull distribution fit into this scheme? Is there no integration over wind velocity in the integral?

  That is correct, we use the average freestream wind speed for each wind direction. The wind rose data is collected into wind direction bins with a user-specified width. The frequency of each wind direction is computed as the number of data points within that bin divided by the total number of discrete data points comprising the wind rose. Then, the wind speed for each bin is the mean of the wind speed of each data point that falls within the wind direction bin. We do not fit a Weibull distribution to the wind speeds within each wind direction bin, so there is no integration over wind speed. That being said, allowing for a distribution of wind speeds is a component of future work.

- Line 96. says "the product is a vector with length equal to the number of wind direction bins", however, this will have units of m/s so how can it be a number? Please clarify.

  We have adjusted the sentence to avoid confusion. We define $g(\theta') = \frac{1}{2\pi}U_\infty(\theta')f(\theta')$ and $h(\theta') = \frac{1}{2\pi}\left[1 - \sqrt{1 - C_T(U_\infty(\theta'))}\right]U_\infty(\theta')f(\theta')$.

- Line 100. for a given g(theta') and N.

  $N$ has been added to be more complete:

  "The Fourier coefficients $a_0$, $a_n$, $b_n$, $c_0$, $c_n$, and $d_n$ can be easily found for a given $g(\theta')$, $h(\theta')$, and $N$."

- Line 102. Taylor series to second order?

  We clarify this in the manuscript:

  "Also, we approximate the fraction in the second term in the right hand side of Eq. 6 using a second-order Taylor expansion"

- It is not clear to me why the integral is intractable (line 121).

  The reason for the integral to be intractable is because unresolved terms that appear when taking the mean. If we write $U = \overline{U} + u'$, where $\overline{U}$ is the wind direction average and $u'$ would be the variations about that average, the average of the cube of the velocity would result in: $\overline{U^3} = \overline{U}^3 + 3\overline{U}\overline{u'^2} + \overline{u'^3}$ . The approximation that we propose neglects the cross terms $3\overline{U}\overline{u'^2} + \overline{u'^3}$.

4. AEP Comparison

- Could the authors briefly state how AEP is calculated? Typically this would be done with a wind rose/Weibull distribution and a wind turbine power curve? Is this the case here?

  A definition for the numerical AEP calculation has been added to the introduction:

  "The AEP of the wind plant is therefore a numerical integral of the power as a function of wind speed and direction:
  $$AEP = \sum_{i=1}^{WD}\sum_{j=1}^{WS} P(\theta'_i, U_\infty(\theta'_i)_j)f_i f_j,$$

where $P$ is the total power of the wind farm as a function of wind direction $\theta'$ and freestream wind speed $U_\infty$ and $f_i$ and $f_j$ are the frequency of each discrete wind direction and speed, respectively."

- I think the various approaches need to be defined clearly and consistent terminology used throughout the paper. For example, Conventional Jensen approach = Jensen wake + numerical integration; FLOWERS = analytical formula; Conventional Gaussian approach = Gaussian wake + numerical integration; It seems sometimes the first is referred to as 'Jensen integration' (line 183) or 'conventional numerical integration method' (line 192) or 'numerical integration' (line 198).

  This is a good point. We have modified the text and now refer to "FLOWERS", "Jensen numerical integration," and "Gauss numerical integration" more consistently throughout the manuscript.

- Line 170. Is the wind rose specified by f(theta')?

  $f(\theta')$ is defined as the frequency of each wind direction bin specified by $\theta'$. The other relevant data for each wind rose is the average freestream wind speed $U_\infty(\theta')$. In Section 3.1 and Section 4, the freestream wind speed is a uniform 8 m/s for every wind direction, but in general the wind rose is represented by a unique wind speed and frequency for each wind direction bin.

- Figure 2. caption says AEP comparison, but it seems that it is the wind velocity that is shown.

  The caption has been corrected:

  "Annually-averaged flow field comparison for three turbines aligned with the predominant wind direction with $6D$ spacing."

- It is stated that "wind direction bins B used in (c) is B = 72", but this is for a single wind direction as shown in subfigure (a), is that right? Does this mean the single wind direction is split into 72?

  The entire wind rose is split into 72 wind direction bins. However, only 5 of these discrete wind directions ($260°\pm2.5°$, $265°\pm2.5°$, $270°\pm2.5°$, $275°\pm2.5°$, $280°\pm2.5°$) have a frequency above 1%. The other wind directions are not visible on the wind rose in Figure 2a because their frequency is negligible.

- Can you give an indication of the absolute computational times? Minutes, hours, days?

  These calculations take about 0.1 - 1 s. This context has been added to the figure caption (see next answer).

- Figure 5. Please state what they values are normalised with respect to. Is it with respect to the cases with maximum resolution?

  We have clarified that the data is normalized by the highest-resolution case in each plot. The new caption is as follows:

  "The effect of the resolution of wind direction bins in the Jensen model (left) and number of Fourier modes in FLOWERS (right) on cost (top) and accuracy (bottom) of the AEP calculations. Each colored line represents one of five sampled wind roses. AEP is computed for a 7-by-3 grid of turbines with a spacing of $5D$. Values are normalized in each plot by the highest-resolution data point. For context, computation times are on the order of seconds."

- Line 225. Perhaps replace "There is no reason to use the extended Fourier series if it only increases the computational cost of the FLOWERS solution" with "There is no reason to use the extended Fourier series if it increases the computational cost of the FLOWERS solution with no associated accuracy benefit."

  This sentence has been revised in the manuscript:

  "There is no reason to use the extended Fourier series if it increases the computational cost of the FLOWERS solution with no benefit to accuracy."

5. Optimization Comparison

- Figure 7. Are labels (a), (b) necessary?

  The subfigure labels are required by the journal.
  From `https://publications.copernicus.org/for_authors/manuscript_preparation.html#figurestables`: "Labels of panels must be included with brackets around letters being lower case (e.g. (a), (b), etc.)." .

- Figure 7. It is a bit concerning that the two results are vastly different, but possibly due to the fact that a top-hat wake is used instead of the more realistic gaussian wake. It is more reassuring that the layouts in Figure 10 are more similar. Also, the initial positions are marked. Surely the final optimised layout should be independent of the initial positions?

  Yes, the FLOWERS and Jensen solutions are qualitatively different. We can attribute this to the discrete boundaries of the wake in the top-hat model, while the FLOWERS approach involves continuous wake profiles similar to the Gaussian model. The final layouts are in fact sensitive to the initial positions in this example. We present a simple test case with a single predominant wind direction, meaning that there are many local optimal solutions that can achieve sufficient wake avoidance. With a much smaller convergence threshold for the optimization study, it is possible that a global optimal solution could be obtained consistently. However, we use a less strict convergence threshold to obtain results in a quicker turnaround time (on the order of hours rather than days for each case) that can still illustrate the key features of our new model. This issue would likely not arise for a wind rose with a more realistic distribution of wind directions that does not permit such a large number of local optimal solutions. We add this sentence to the manuscript to reinforce this idea:

  "Finding a global optimum in this unrealistic test case would require a stricter convergence criterion. We note that a more realistic wind rose would likely not enable so many local optima in the solution space, so this artifact should not be as pronounced in practice."

- Line 264. Please define AEP gain.

  A definition for AEP gain has been added to the manuscript:

  "The metric to compare the quality of optimal solutions is AEP gain:

  $$G_{AEP} = \frac{AEP_{opt} - AEP_{init}}{AEP_{init}} * 100\%,$$

  where $AEP_{init}$ is the AEP of the initial layout and $AEP_{opt}$ is the AEP of the final solution."

6. Conclusions: As future work it would be of interest to validate the AEPs of the various approaches against a real wind farm AEP. Figures 8, 9, 11, 12 don't actually indicate which method is actually closest to the true AEP of a real wind farm.

   Validation of the AEP calculation through FLOWERS could be a component of future work. Validation of AEP from other low-fidelity wake models has already been performed extensively in the literature. We clarify that we are less interested in the accuracy of the AEP estimate from FLOWERS than in the ability for FLOWERS to achieve optimal plant layouts using the new formulation for AEP. So, LES validation of the performance of the optimal layouts produced by FLOWERS versus the other methods is a key component of our future work, rather than validating the estimate of AEP itself. The following sentecne has been adjusted in the conclusions: "We also plan to validate the performance of the FLOWERS optimal solutions with high-fidelity simulations"

**RC2**

The authors propose to apply an analytical integration method for evaluating the annual energy production (AEP) of a wind farm and compare its performance against the more common numerical integration method. The article is well written and contains novel ideas. However, I do have some major concerns with the assumptions of proposed method and the employed methodology for comparing the analytical and the

traditional AEP integration methods. More detailed comments are listed below;they need to be addressed before the article can be considered for publication in Wind Energy Science.

We thank the reviewer for their positive comments, attention to detail, and constructive feedback. The responses to the comments are below and marked in blue.

1. My main concern with this work is that one needs to use constant thrust and power coefficients (for all wind speeds) in order to be able to perform an analytical integration of the AEP by integrating the velocity deficits for all wind directions and wind speeds. Wind turbines do not have constant thrust and power coefficients above rated wind speed, which makes the proposed method difficult to use for wind farm layout design and optimization. The authors propose to include more realistic thrust and power coefficients in future work, but I not see how this can be done, since the current derivation of the analytical model only holds for constant thrust and power coefficients. In addition, the errors from using constant thrust and power coefficients for a wind farm layout optimization are not properly addressed because you only investigate a single below rated wind speed where the assumption is not violated significantly. Therefore, you need to add a realistic case as well (including all wind speeds, for example from 4-25 m/s) to access the errors associated to the constant thrust and power coefficients in terms of AEP.

This is a valid point. We can allow for a wind-speed dependent $C_T$ and $C_p$ by slightly modifying our derivation. We have changed the derivation to include this result:

"We define $g(\theta') = \frac{1}{2\pi} U_\infty(\theta') f(\theta')$ and $h(\theta') = \frac{1}{2\pi} \left[ 1 - \sqrt{1 - C_T(U_\infty(\theta'))} \right] U_\infty(\theta') f(\theta')$, which are both vectors with length equal to the number of wind direction bins. If we expand $g(\theta')$ and $h(\theta')$ with Fourier series:

$$g(\theta') = \frac{c_0}{2} + \sum_{n=1}^{N} c_n \cos(n\theta') + d_n \sin(n\theta')$$

$$h(\theta') = \frac{a_0}{2} + \sum_{n=1}^{N} a_n \cos(n\theta') + b_n \sin(n\theta'),$$

then the wind rose is defined continuously rather than discretely. The Fourier coefficients $a_0$, $a_n$, $b_n$, $c_0$, $c_n$, and $d_n$ can be easily found for a given $g(\theta')$, $h(\theta')$, and $N$."

Then, the solution for $\overline{U(r,\theta)}$ is computed using the Fourier expansion of these two separate functions:

$$\overline{U(r,\theta)} = \int_{-\pi}^{\pi} g(\theta')\, d\theta' \quad - \sum_i \int_{-\theta_c}^{\theta_c} h(\theta_i - u) \left[ \frac{1}{(kr+1)^2} + \frac{kru^2}{(kr+1)^3} \right] du.$$

The difference between this approach and the conventional method is also clarified in the manuscript:

"In FLOWERS, we neglect local wind conditions when computing the wake velocity and instead normalize the velocity deficit by the global freestream velocity and assume a uniform $C_p$ and $C_T$ across all turbines; the Jensen numerical integration implementation, on the other hand, normalizes the velocity deficit with and computes $C_p$ and $C_T$ as a function of the local wind velocity."

In our formulation, $C_p$ is treated as a function of the average wake speed $\overline{U}$. With this approach, we must still assume that $C_T$ and $C_p$ are uniform across the wind farm. Allowing the coefficients to vary among turbines requires implementing local flow conditions into the derivation, which is a part of future work. Using $C_T(U_\infty)$ and $C_p(\overline{U})$ instead of constant $C_T$ and $C_p$ will capture the variations in the coefficients in domains of low and high wind speeds. We have incorporated this change into the derivation of the model and results.

We stand by the simple test case used to compare the new modeling approach with the conventional methods with a single freestream wind speed and predominant wind direction. This paper's main objective is to outline the derivation of the model and illustrate preliminary results for its performance against the conventional approach. Validation of the FLOWERS optimal solutions with high-fidelity simulations and applications to more realistic wind roses and farm boundaries is the subject of future work.

2. The motivation of this work is to speed up an AEP calculation using an engineering wake model. The authors report a calculation time of 16700 s for a wind farm layout optimization using a numerical AEP integration within FLORIS using a wind farm of 9 wind turbines, 360 wind directions and a single wind speed. It is not clear how many different wind farm layouts are evaluated within the optimization process, but even if this was 1000 then the calculation time per layout seems to be quite long considering the small number of turbines and the use of only a single wind speed. In PyWake [2], a single AEP calculation for a larger wind farm (e.g. 80 turbines) using 360 wind directions and 22 wind speeds typically takes less than a second, so one could easily perform a wind farm layout optimization using a numerical integral. I suspect that the numerical implementation in FLORIS could be improved significantly.

An expected number of layouts to evaluate within one optimization study is on the order of 1000. To calculate the AEP of the 21 turbine wind farm in Section 3.3 (Figure 5), the FLORIS implementation with 5° wind direction bins requires about 1 second; so, with 360 wind directions, we could expect a computation time of about 5 s for this size wind farm.

As you point out, this time is slower than what you cite for PyWake. We acknowledge that the current implementation of the numerical integration approach within FLORIS uses non-vectorized calculations, which adds computational cost from the use of for-loops. We intend to compare to a vectorized code in our future work–perhaps we could implement FLOWERS within PyWake to compare performance. We have also added a note along these lines to the manuscript:

"We should note that the implementation of the Jensen numerical integration in FLORIS Version 2 is non-vectorized (i.e. calculations are mostly performed through for-loops instead of vector operations). Comparisons in computation time with a vectorized code such as PyWake would likely show a smaller discrepancy. However, the code implementation of FLOWERS is not fully vectorized either, so it is a fair comparison of FLOWERS to the Jensen numerical integration to showcase the performance of our newly-developed model."

We clarify that the objective of this paper is not a detailed comparison between softwares, but rather a comparison of our newly-proposed modeling framework to a conventional implementation of AEP calculations and layout optimization programs to illustrate the potential for computational savings.

3. The authors mention that the analytical integration method is more smooth and therefore better suited for optimization. While this is indeed true, one could also consider a numerical integration method using the Gaussian wake model including analytical functions for all the discrete input (Weibull, wind rose, power and thrust coefficient curve), see for example [1]. (This reference also investigates how many flow cases are necessary for an AEP calculation, which you also look at, so you could cite it.)

We agree that using analytic functions for discrete inputs such as the wind rose and power and thrust coefficients does save time versus using finite differences. However, since the underlying inputs to those analytic functions are still discrete, the issue of a discrete number of points that create local minima in the solution persists. The Fourier approximation is the key to avoiding this unwanted behavior by representing the discrete input data as a continuous function. We have added a citation to the referenced paper in the manuscript:

"Research that addresses the calculation of AEP has focused on statistical methods to improve the efficiency of estimating this quantity (King et al., 2020; Padrón et al., 2019), or on defining the discrete inputs of the wake model (such as the wind rose and power and thrust curves) with analytical functions (Murcia et al., 2015)."

4. Figure 5: It is interesting to see that the analytical integration works well for only a small number of Fourier modes.

Thank you for the positive remark.

5. Your definition of the wake region does not hold because it would imply that there is a wake region upstream (for negative values of x). So you would need to be more precise.

Thank you for spotting that error. This text has been clarified in the manuscript:

"$W(x, y)$ represents the Jensen wake region: $W(x, y) = 1$ if $|y| \leq kx + 1$ and $x \geq 0$ and is zero elsewhere."

6. Equation (1): You forgot to define the origin, I guess it is the wind turbine location. In addition, the original NOJ model use $1 + 2k$ so your wake expansion coefficient is twice as large as the standard model. I think you should mention this to avoid confusion.

This is a good point, and the definition has been clarified in the manuscript:

"...$x$ and $y$ are the streamwise and spanwise position, respectively, normalized by the rotor radius with the origin at the turbine location." For the second point, if you normalize the downstream distance by the rotor diameter, which we believe the original model does, then the 2 will be present as you note. However, since we normalize $x$ by the rotor radius, we maintain consistency:

$$\frac{U(x, y)}{U_\infty} = 1 - \frac{1 - \sqrt{1 - C_T}}{(kx/R + 1)^2} W(x, y) = 1 - \frac{1 - \sqrt{1 - C_T}}{(2kx/D + 1)^2} W(x, y)$$

7. Sections 3.1 and 3.2: A difference of 1.8% and 8% in AEP is quite a lot considering the amount of money 1% AEP represents in a large wind farm. Therefore, a wind farm developer would go for the more precise numerical method instead of the analytical method if the error is that large.

This is true, but we are not claiming that the AEP estimate from FLOWERS is accurate. We demonstrate that even with differences in the AEP estimate compared with the numerical integral, we can still achieve similar performance between the two optimizers in terms of AEP of the final layout. We use the conventional method to compute AEP for all of the cited AEP values in Section 4 so that there is no added discrepancy from the FLOWERS AEP–the difference is the AEP method that is implemented in the objective function within the layout optimizer.

Also, we must recognize that any of these low-fidelity approximations for AEP are likely inaccurate. They deal with large levels of uncertainty and ultimately a wind farm developer would want to use a high-fidelity simulation to verify the expected AEP of the plant before construction. That being said, improving the accuracy of the FLOWERS formulation is a subject of future work, mainly by accounting for local wind conditions and integrating a Gaussian model instead of a top-hat model.

8. Section 4.1: When you compare the results of optimized AEP between FLOWERS and Jensen, do you post evaluate the AEP of the optimized layout from FLOWERS using the numerical approach? If not, then you cannot really compare the AEP values. In addition, why do you use wake added turbulence for the Jensen model and not for the FLOWERS model? Wouldn't be more fair to apply the Jensen model in the same way (so without added wake turbulence)?

Yes, we calculate AEP for the optimal solutions using the numerical approach (stated in the beginning of Section 4). We do not include wake-added turbulence in FLOWERS because we focus on the simplest implementation of the new model formulation for this first paper. Wake-added turbulence could be a component of future work for this model, just as the consideration of local flow conditions is. We want to compare FLOWERS against the most common version of the numerical integration code to assess expected performance, which would include wake-added turbulence. The following sentence has been added to the manuscript to clarify this reasoning:

"We have neglected wake-added turbulence in the FLOWERS formulation to present the simplest form of this new model, but we compare against the Jensen numerical integration that includes wake-added turbulence because we expect this effect to be modeled in most wake modeling codes."

9. Can you use your analytical AEP integration model if the wake superposition method is nonlinear? The wake superposition model is important part of AEP calculation using engineering wake models and one might typically want to aplly several wake superposition methods in order to get an estimate of the model uncertainty.

   The current formulation relies on a linear superposition method for the integral to be analytically tractable. There is research suggesting that the linear method is more accurate, especially with a Gaussian model (which we plan to implement in the future). Uncertainty quantification techniques are a potential area of exploration within this framework down the road. We have added this sentence to the conclusions to foreshadow our future work:

   "Future work will also focus on studying the effects of superposition methods and model uncertainty in the FLOWERS formulation."

10. The title could be more clear. When I first read it I did not think of an analytic AEP integration method.

    We have changed the title to:

    "FLOW Estimation and Rose Superposition (FLOWERS): An integral approach to engineering wake models".

---

## Author Response (AR2)

**FLOWERS Reviewer Comments II**

Michael J. LoCascio et al.

April 21, 2022

**Comment 1**

"Thank you for your corrections. I agree with most of the changes but I find it a missed opportunity to not report the difference between the analytical and the numerical approaches for a realistic AEP calculation (using a wind speed range of 4-25 m/s instead of just a single below rated wind speed), as pointed out in my first main comment. If I use PyWake to calculate the AEP of Horns Rev II using the standard setup as reported in my first review and an equivalent setup using a constant CT=0.8 and CP=0.44 for all wind speed then I get: AEP = 1096.37 GWh instead of 702.43515771906 GWh, which is an error of 56%. In terrm of wake loss the error is about 100%. Therefore, I think you need to report the difference in AEP for a realistic AEP case and not just a single wind speed below rated wind speed (where the constant CT and CP assumption is not tested properly)."

We appreciate the comment from the reviewer. We have added a more realistic AEP calculation as requested for a wind rose with higher wind speeds on a larger wind farm. The following text is now in the manuscript:

"For a more realistic wind rose, the discrepancies in AEP between FLOWERS and the Jensen numerical integration approach are more substantial. We consider a larger wind farm of sixty turbines with $6D$ spacing in the streamwise and spanwise directions and an offshore wind rose sampled from the WIND Toolkit draxl2015wind with higher average freestream wind speeds than those considered in the previous example. Figure 1 displays the resulting flow fields from the FLOWERS and Jensen integration models. The AEP computed with FLOWERS is about 17% lower than that from the Jensen model. This greater difference between the two approaches in this case can be attributed to the assumption that $C_T$ and $C_P$ do not depend on the local flow velocity at each turbine, which breaks down for larger wind farms with more heterogeneity in the flow velocity. Including the local flow velocity in the FLOWERS formulation is part of future work."

We also clarify the treatment of $C_T$ and $C_P$. After our previous revisions based on the reviewer's first round of comments, $C_T$ varies with the freestream wind speed, which is itself a function of wind direction: $C_T(U_\infty(\theta'))$. This definition means that $C_T$ is the same for every turbine in the wind farm for a particular wind direction but still varies across different wind directions and speeds. $C_P$ is computed as a function of the average wake speed (i.e. it does not vary for different wind directions): $C_P(\overline{U})$. While this definition of $C_P$ is an approximation and does not exactly account for its dependence on the local flow velocity for different wind directions, it does incorporate the effect of wake velocity deficits on power production (since we use the wake speed and not the freestream wind speed). Both $C_T$ and $C_P$ are not defined as arbitrary constants—they do include the variation in wind speed over a realistic wind rose because both are computed as a function of $U_\infty(\theta')$ (directly in the case of $C_T$ and indirectly via the average wake speed $\overline{U}$ in the case of $C_P$).

The following two sentences have been modified to make this point as clearly as possible:

- "We make two simplifications to compensate: $C_p$ is calibrated as a function of the average wake speed $\overline{U}$ (which for each turbine is a constant across all wind directions), and we substitute $\overline{U}^3$ into Eq. 13 instead of evaluating $\overline{U^3}$."

- "$C_T$ is the thrust coefficient which is realistically a function of the local inflow speed; we simplify it to be a function of the freestream wind speed $U_\infty$ such that it is uniform across all turbines in the wind farm but still varies around the wind rose."

[Figure]

Figure 1: Flow field comparison for a sixty turbine wind farm with $6D$ spacing with a realistic wind rose (corresponding to an offshore site in the Northwest U.S.). Again, the number of wind direction bins $B$ in the numerical method (c) is $B = 72$ , and the number of Fourier coefficients in FLOWERS (b) is $N = 37$.

**Comment 2**

Minor point: What does s.t. stand for in equation (14). Is it 'subject to'? If so perhaps replace s.t. with subject to.

Thank you for pointing this wording out—the language of the equation is unclear. We have modified the equation to state that we are optimizing $AEP$ subject to certain constraints:

$$\min_{x_i, y_i} \quad -AEP\left(x_i, y_i, U_\infty(\theta'), f(\theta')\right)$$
$$\text{subject to} \quad \text{boundary constraints}$$
$$S_{ij} \geq 2D,$$